# Prostate-specific antigen dynamics predict individual responses to intermittent androgen deprivation

Renee Brady-Nicholls [1], John D. Nagy[2,3], Travis A. Gerke[4], Tian Zhang[5], Andrew Z. Wang [6], Jingsong Zhang[7], Robert A. Gatenby[8✉] & Heiko Enderling [1✉]

Intermittent androgen deprivation therapy (IADT) is an attractive treatment for biochemically recurrent prostate cancer (PCa), whereby cycling treatment on and off can reduce cumulative dose and limit toxicities. We simulate prostate-specific antigen (PSA) dynamics, with enrichment of PCa stem-like cell (PCaSC) during treatment as a plausible mechanism of resistance evolution. Simulated PCaSC proliferation patterns correlate with longitudinal serum PSA measurements in 70 PCa patients. Learning dynamics from each treatment cycle in a leave-one-out study, model simulations predict patient-specific evolution of resistance with an overall accuracy of 89% (sensitivity = 73%, specificity = 91%). Previous studies have shown a benefit of concurrent therapies with ADT in both low- and high-volume metastatic hormone-sensitive PCa. Model simulations based on response dynamics from the first IADT cycle identify patients who would benefit from concurrent docetaxel, demonstrating the feasibility and potential value of adaptive clinical trials guided by patient-specific mathematical models of intratumoral evolutionary dynamics.

[1] Department of Integrated Mathematical Oncology, H. Lee Moffitt Cancer Center and Research Institute, 12902 USF Magnolia Drive, Tampa, FL 33612, USA. [2] Department of Life Sciences, Scottsdale Community College, 9000 E. Chaparral Rd., Scottsdale, AZ 85256, USA. [3] School of Mathematical and Statistical Sciences, Arizona State University, 900 S Palm Walk, Tempe, AZ 85281, USA. [4] Department of Cancer Epidemiology, H. Lee Moffitt Cancer Center and Research Institute, 12902 USF Magnolia Drive, Tampa, FL 33612, USA. [5] Division of Medical Oncology, Department of Medicine, Duke Cancer Institute, 20 Duke Medicine Cir, Durham, NC 27710, USA. [6] Department of Radiation Oncology, University of North Carolina at Chapel Hill, Chapel Hill, NC 27599, USA. [7] Department of Genitourinary Oncology, H. Lee Moffitt Cancer Center and Research Institute, 12902 Magnolia Drive, Tampa, FL 33612, USA. [8] Department of Radiology, H. Lee Moffitt Cancer Center and Research Institute, 12902 Magnolia Drive, Tampa, FL 33612, USA. ✉email: Robert.gatenby@moffitt.org; heiko.enderling@moffitt.org

Prostate cancer (PCa) is the most common type of cancer in American men and the second leading cause of cancer mortality[1]. Following surgery or radiation, the standard treatment for hormone-sensitive PCa is continuous androgen deprivation therapy (ADT) at the maximum tolerable dose (MTD) with or without continuous abiraterone acetate (AA) until the tumor becomes castration resistant[2]. Importantly, advanced PCa is not curable because PCa routinely evolves resistance to all current treatment modalities. Continuous treatment approaches fail to consider the evolutionary dynamics of treatment response where competition, adaptation, and selection between treatment sensitive and resistant cells contribute to therapy failure[3]. In fact, continuous treatment, by maximally selecting for resistant phenotypes and eliminating other competing populations, may actually accelerate the emergence of resistant populations—a well-studied evolutionary phenomenon termed competitive release[4].

In part to address this issue, prior trials have used intermittent ADT (IADT) to reduce toxicity and potentially delay time to progression (TTP). However, these trials were typically not designed with a detailed understanding of the underlying evolutionary dynamics. For example, a prospective Phase II trial of IADT for advanced PCa included an 9-month induction period in which patients were treated at MTD prior to beginning intermittent therapy[5]. We have previously postulated that only a small number of ADT-sensitive cells would typically remain after the induction period, thereby significantly reducing the potential of intermittent treatment to take advantage of the evolutionary dynamics[3].

Fully harnessing the potential of intermittent PCa therapy requires identifying ADT resistance mechanisms, predicting individual responses, and determining potentially highly patient-specific, clinically actionable triggers for pausing and resuming IADT cycles. Progress in integrated mathematical oncology may make such analysis possible. Many mathematical models based on a variety of plausible resistance mechanisms have been proposed to simulate IADT responses[3,6–14]. Although these models can fit clinical data, they often rely on numerous model variables and parameters that in combination fail to adequately predict responses and outcomes for individual patients[10]. We hypothesize that PCa cells with stem-like properties (PCaSCs) may be, at least in part, responsible for tumor heterogeneity and treatment failure owing to their self-renewing, differentiating and quiescent nature[15–17]. Here, we define PCaSCs to be a population of cells that are less sensitive to low testosterone environments but still somewhat dependent on androgen receptor pathways. Simulating longitudinal prostate-specific antigen (PSA) levels in early IADT treatment cycles could help identify patient-specific PCaSC dynamics to computationally forecast individual disease dynamics and reliably predict IADT response or resistance in subsequent treatment cycles.

The first evidence of stem cells in the prostate was provided by Isaacs and Coffey[18], who used androgen cycling experiments in rodents to show that castration resulted in involution of the prostate, whereas restored androgen levels resulted in complete regeneration of the prostate. These findings demonstrated that the normal prostate depends on androgens for maintenance. A small population of androgen-independent stem cells within the prostate epithelium divide to give rise to amplifying cells, which do not directly depend on androgen for their continuous maintenance, but respond to androgens by generating androgen-dependent transit cells. Approximately 0.1% of cells in prostate tumors express the stem cell markers $CD44^+/\alpha2\beta1^{hi}/CD133^+$[19]. A pre-clinical study by Bruchovsky et al. showed ADT selects for murine PCaSCs[20]. Analogously, Lee et al.[21] demonstrated increased PCaSCs populations after ADT in patient-derived PCa

cell lines, which can be reverted by the addition of functional AR. Combined, these results suggest evolution of or selection for pre-existing androgen-independent PCaSCs as a plausible explanation of the development of ADT resistance.

The purpose of this study is to evaluate individual PSA dynamics in early IADT treatment cycles as a predictive marker of response or resistance in subsequent treatment cycles. We hypothesize that patient-specific PCaSC division patterns underlie the measurable longitudinal PSA dynamics, and that a mathematical model of PCaSCs can be trained to predict treatment responses on a per-patient basis. Here, we present an innovative framework to simulate and predict the dynamics of PCaSCs, androgen-dependent non-stem PCa cells (PCaCs), and blood PSA concentrations during IADT. Our mathematical model of PCaSC enrichment is calibrated and validated with longitudinal PSA measurements in individual patients to identify model dynamics that correlate with treatment resistance. The model's predictive power to accurately forecast individual patients' responses to IADT cycles is evaluated in an independent patient cohort. These analyses suggest that PCaSC and PSA dynamics may potentially be used to personalize IADT, maximize TTP, and ultimately improve PCa outcomes. The calibrated and validated model is then used to generate testable hypotheses about patients that may benefit from concurrent chemotherapy.

## Results

**Model accurately simulates patient-specific response dynamics.** The model was calibrated to and assessed for accuracy on longitudinal data from a prospective Phase II study trial conducted in 109 men with biochemically recurrent PCa treated with IADT[22] (see Methods, Supplementary Fig. 1). Stratified random sampling[23] was used to divide the data into training and testing cohorts. That is, the data were ordered according to number of cycles of treatment and then randomly sampled such that each cohort had a similar spread of number of cycles per patient. Assuming that uninhibited PCaSCs divide approximately once per day[24], $\lambda$ (day$^{-1}$) was set to ln(2) and parameter estimation was used to find the remaining parameters. The model was calibrated to the training cohort data with two population uniform parameters (PSA production rate $\rho = 1.87\text{E-}04$ (µg/L day$^{-1}$), decay rate $\varphi = 0.0856$ (day$^{-1}$)) and two patient-specific parameters (median PCaSC self-renewal rate $p_s = 0.0278$ (95% CI: [0.0265, 0.0611]) (non-dimensional), ADT cytotoxicity rate $\alpha = 0.0360$ (95% CI: [0.0258, 0.1379]) (day$^{-1}$)).

The model results captured clinically measured longitudinal PSA dynamics of individual responsive and resistant patients (Supplementary Fig. 2A–B) and the population as a whole ($R^2 = 0.74$ Supplementary Fig. 2C). The corresponding PCaSC dynamics demonstrated a rapid increase in the PCaSC population in patients that became castration resistant compared with patients that remained sensitive throughout the trial. Simulations also showed that, in this model, the emergence of resistance is a result of selection for the PCaSCs during on-treatment phases. Coinciding with this transition to castration resistance, analysis of model parameters revealed that resistant patients had significantly higher PCaSC self-renewal rates than responsive patients (median $p_s = 0.0249$ for responsive patients vs. $p_s = 0.0820$ for resistant patients, $p < 0.001$, (95% CI: (0.0167, 0.0344) vs. (0.0472, 0.1460), respectively) Supplementary Fig. 2D). As such, we can conclude that those patients with a low $p_s$ are considered to be evolutionarily stable, whereas those with a higher $p_s$ are likely to develop resistance quicker. As shown in Supplementary Fig. 2D, analysis revealed that $p_s$ and $\alpha$ may be functionally related. Supplementary Table 1 shows the mean squared error

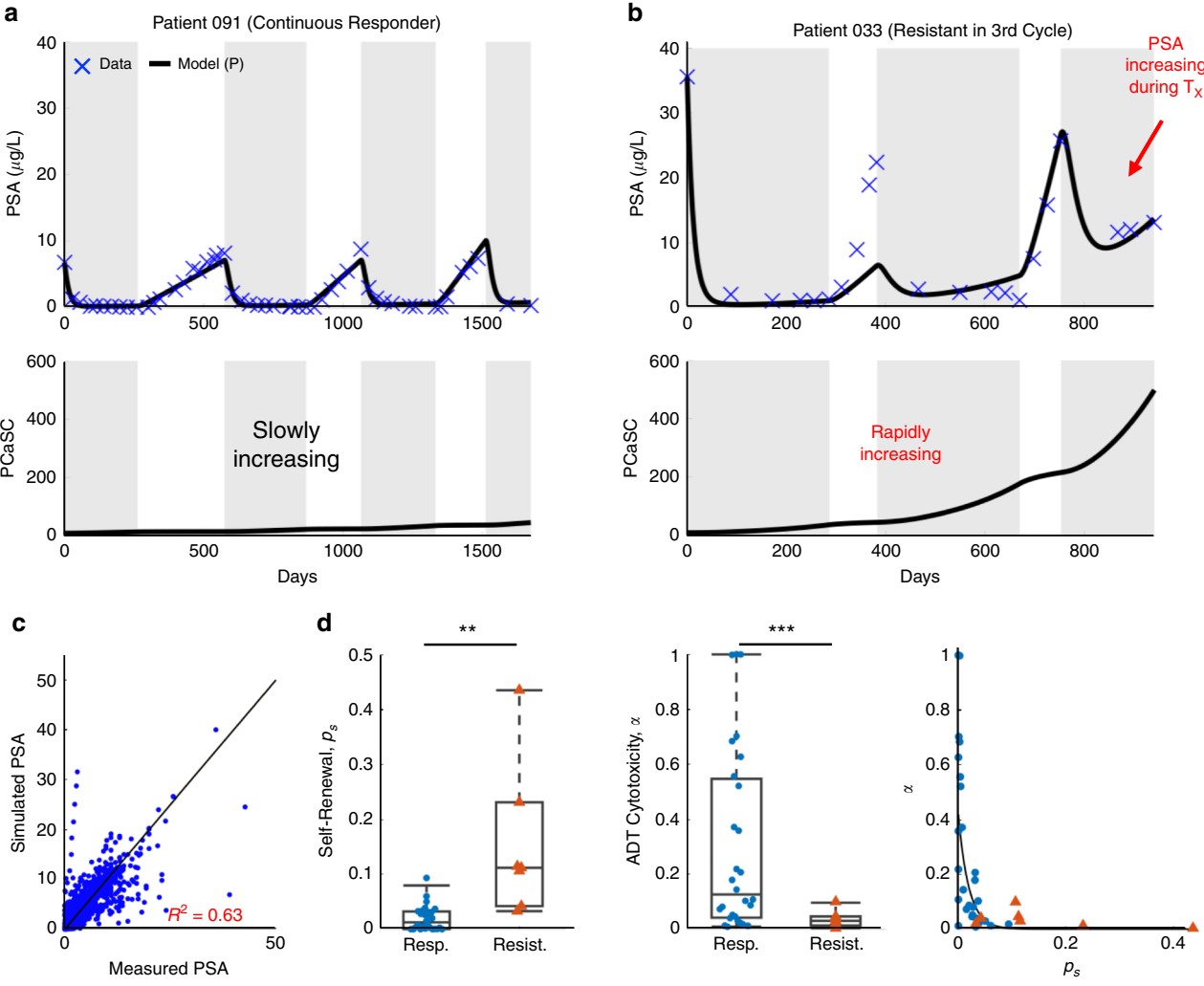

**Fig. 1 Model validation on testing patients. a**, **b** Model fits to PSA data and corresponding PCaSC dynamics for **a** a continuous responder and **b** a patient who developed resistance during his third cycle of treatment. PCaSC population is rapidly increasing in resistant patient and slowly in responsive patient owing to a significantly higher self-renewal rate ($p_s = 0.0201$ and $0.1118$ for patients 091 and 033, respectively). **c** Simulated vs. measured PSA. Linear regression obtains an R2 of 0.63. **d** Parameter distributions for the stem cell self-renewal $p_s$ and ADT cytotoxicity $\alpha$, comparing between responsive ($n = 28$) and resistance ($n = 7$) patients, with $\varphi$ and $\rho$ learned from training patients. Boxplots show median including the 25th and 75th percentiles. The two-sample $t$ test was used to calculate the statistical significance of the difference between the two groups ($p = 0.016$ (left pane), $p = 2.54\text{E-}05$ (right pane), significance level denoted by double and triple star, respectively). Exponential relationship between ps and $\alpha$. Blue dots and red triangles denote responsive and resistant patients, respectively.

comparison for possible relationships between these parameters. This analysis showed that multiple relationships could provide similar fits to the data. From this, we chose the exponential relationship, allowing PCaSC self-renewal rate $p_s$ to be the single, identifiable, independent patient-specific parameter.

To assess the accuracy of our model, we set the PSA production rate $\rho$ and decay rate $\varphi$ to the values obtained from the training cohort and identified patient-specific values for $p_s$ and corresponding $\alpha$ in the testing cohort. With these, the model was able to fit the data ($R^2 = 0.63$) and the resulting parameter distributions and relationships were similar to those found in the training cohort (Fig. 1).

**PCaSC dynamics predict subsequent IADT cycle responses**. To predict the evolution of resistance in subsequent treatment cycles, we fit the model to single treatment cycles for patients in the training set, again setting the PSA production rate $\rho$ and decay rate $\varphi$ to the values previously found (Fig. 2a). The self-renewal

and ADT cytotoxicity rates maintained the exponential relationship, previously obtained when optimizing over all IADT cycles. The distributions of the self-renewal rate $p_s$ and the corresponding ADT cytotoxicity rate $\alpha$, as well as their relative change from cycle to cycle, were used to predict responses in subsequent cycles of patients in the training cohort. To determine whether to categorize a patient as responsive or resistant based on our predictions, receiving operating characteristic curves[25] from the results from the training cohort data were analyzed to find a threshold $\kappa_i$ for each cycle (Fig. 2d). If $P(\Omega) > \kappa_i$, then the patient was predicted as resistant in cycle i. For each cycle i, a value of $\kappa_i$ was chosen that would maximize the sensitivity (predicting resistance when a patient is indeed resistant), and specificity (predicting resistance when a patient is actually responsive) of the training cohort.

Using these thresholds, model forecasting was completed on the testing set and the patient was classified as either responsive or resistant in the subsequent treatment cycle. Representative examples of second and third cycle predictions for one responsive

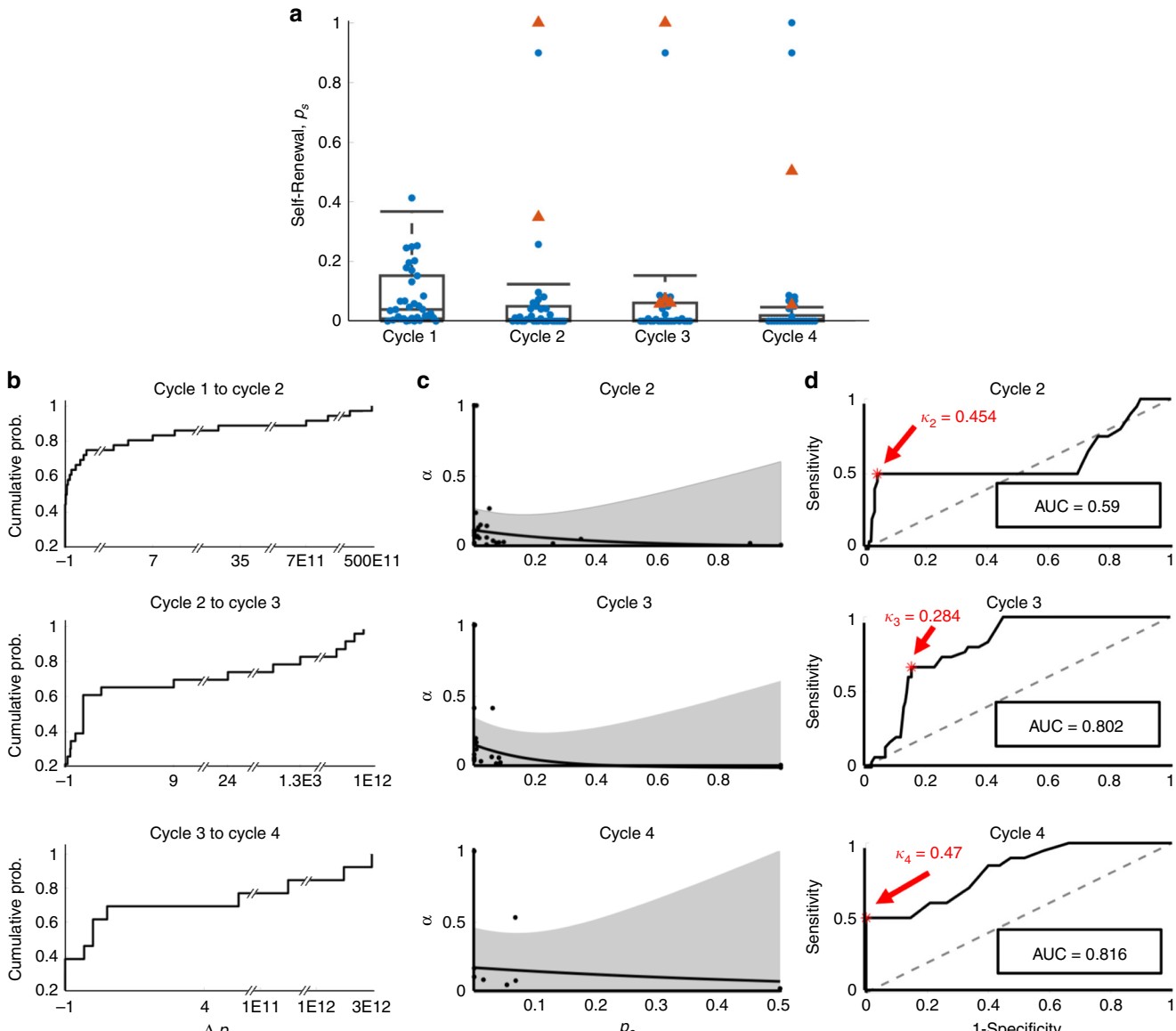

**Fig. 2 Cycle to cycle parameter changes. a** Cycle to cycle parameter distributions for $p_s$ (changes between cycles is not significant ($p > 0.05$)) for $n = 35$, $n = 35$, $n = 33$, $n = 30$ patients in cycles 1 through 4, respectively. Boxplots show median including the 25th and 75th percentiles. The two-sample $t$ test was used to calculate the statistical significance of the difference between the groups. Blue dots and red triangles denote responsive and resistant patients, respectively. **b** Cumulative probability of relative changes in $p_s$ between cycles. **c** $\alpha$ vs. $p_s$ fit (black curve) and 95% confidence interval (gray). **d** Receiver operating characteristic curves for training predictions for cycles two through four. The resistance threshold $\kappa$ that maximizes the accuracy and hence, predictive power, is shown in red.

patient and one resistant patient from the testing cohort are shown in Fig. 3. Patient 017 was a continuous responder who underwent three cycles of IADT before the end of the trial. The model correctly predicted responsiveness in cycles two and three based on the parameters fitted in cycles one and two, respectively. Patient 054 became resistant in the third IADT cycle and the model was able to correctly predict response in cycle two and resistance in the third cycle based on the thresholds learned in the training set. The model yielded a sensitivity of 57% and specificity of 94% over all subsequent IADT cycles for patients in the test cohort. The overall accuracy of the model was 90%.

To evaluate if the low sensitivity is an artifact of which resistant patients are stratified into the training cohort, we performed the parameter optimization as well as forecast analysis by re-randomizing the patients into training and testing cohorts five

times. The 95% confidence intervals for the uniform parameters $\varphi$ and $\rho$, the threshold values $k_i$, as well as the overall accuracy are shown in Supplementary Table 2. As expected, the threshold values varied dependent upon the randomization. The overall accuracy had a relatively narrow 95% confidence interval ranging between 75.34% and 88.66%, with sensitivity between 43% and 61% and specificity between 75% and 91%. To better account for under-represented resistant patients in the training cohort, we utilized a bootstrapping leave-one-out design and performed the optimization and forecast analysis for each of the 70 patients individually. This resulted in training set-specific $\varphi$, $\rho$, $\kappa_2$, $\kappa_3$, and $\kappa_4$ (95% confidence intervals shown in Supplementary Table 3), yielding a sensitivity of 73%, specificity of 91%, and overall accuracy of 89% (mean AUC = $0.80 \pm 0.06$). The ROC curve for each individual patient, as well as the average ROC curve for cycles 2 through 4 are shown in Fig. 4.

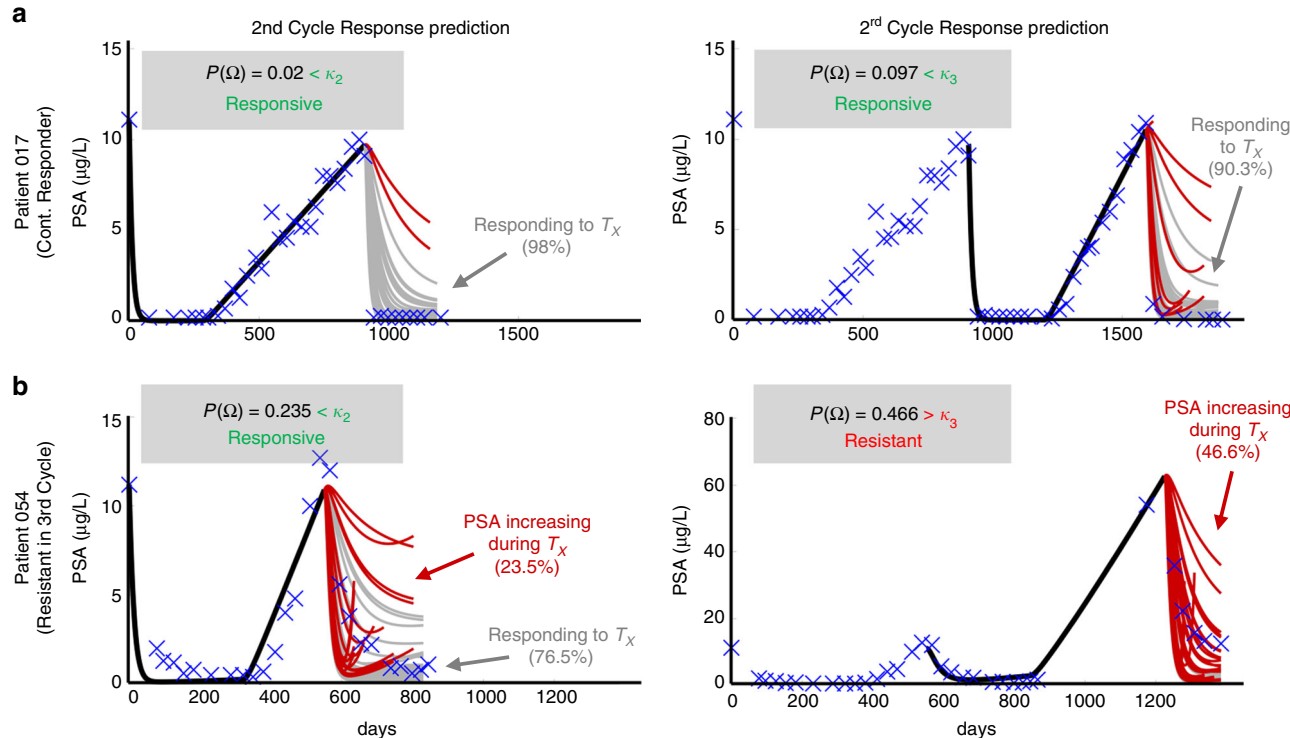

**Fig. 3 Model forecast using parameters obtained by sampling from CDF and 95% confidence intervals. a** The model predicted responsiveness in cycles two (98% of simulations) and three (90.3% of simulations) for Patient 017. The probability of resistant ($P(\Omega)$) was less than its respective $\kappa$ ($\kappa_2 = 0.45$, $\kappa_3 = 0.29$) for each cycle. The patient completed the trial on day = 2202. **b** The model predicted resistance in 23.5% of cycle 2 simulations and in 46.6% of cycle 3 simulations for Patient 054. The predicted probability of resistance in cycle three was greater than $\kappa_3$ so the patient would be advised to stop the trial. Data showed that he became resistant on day 1384 during the third cycle, as predicted.

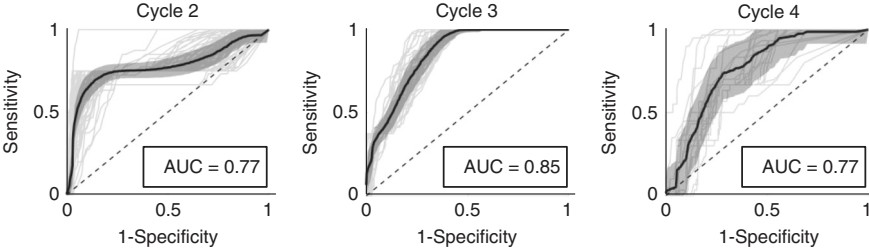

**Fig. 4 Leave-one-out study ROC analysis.** Individual ROC curves for each leave-one-out cohort (gray curves), with mean ROC curve and corresponding 95% confidence interval (black). AUC 95% confidence intervals are (0.76, 0.78), (0.85, 0.86), and (0.74, 0.80), for cycles 2, 3, and 4, respectively.

**IADT without induction period significantly increases TTP.** In a study by Crook et al.[5] continuous ADT was compared with IADT in localized PCa patients and they found that IADT was non-inferior to ADT using overall survival as the clinical end-point. A similar study by Hussain et al.[26] conducted in metastatic, hormone-sensitive PCa patients found that neither regimen proved superior. These findings are likely the result of the 7–8 months induction period, which resulted in the competitive release of the resistant phenotype once the androgen sensitive subpopulation was eliminated[3]. Administering IADT without such an induction period would allow sufficient time for the sensitive subpopulation to efficiently compete with the resistant subpopulation and prolong TTP and ultimately, overall survival.

To test these hypotheses, we simulated continuous ADT and IADT without the 36-week induction period and compared predicted TTP against the trial by Bruchovsky et al.[22] For IADT simulations, treatment was administered until PSA fell below 4

µg/L and resumed again once it rose above 10 µg/L at simulated measurements every 4 weeks. Both protocols were simulated until PSA progression or for 10 years (end of simulation, EOS), with progression defined as three sequential increases in PSA during treatment. For the IADT protocol, progression was also defined as PSA > 4 µg/L at both 24 and 32 weeks on treatment, as used in the Bruchovsky trial. Figure 5a shows the treatment cycling times for the 70 patients included in our analysis. For those that did not progress or experience an adverse event/death before the end of the trial, we continued simulating IADT until progression for up to 10 years (Fig. 5b). Figure 5c–d shows the treatment cycling times from the model simulations of IADT without the induction period and continuous ADT. Without induction, the average cumulative dose is ~88% of that of the standard IADT and ~46% of continuous ADT. Comparing the TTP from the trial results against simulated continuous ADT showed that IADT with an induction period resulted in longer TTP (Fig. 5e), though not

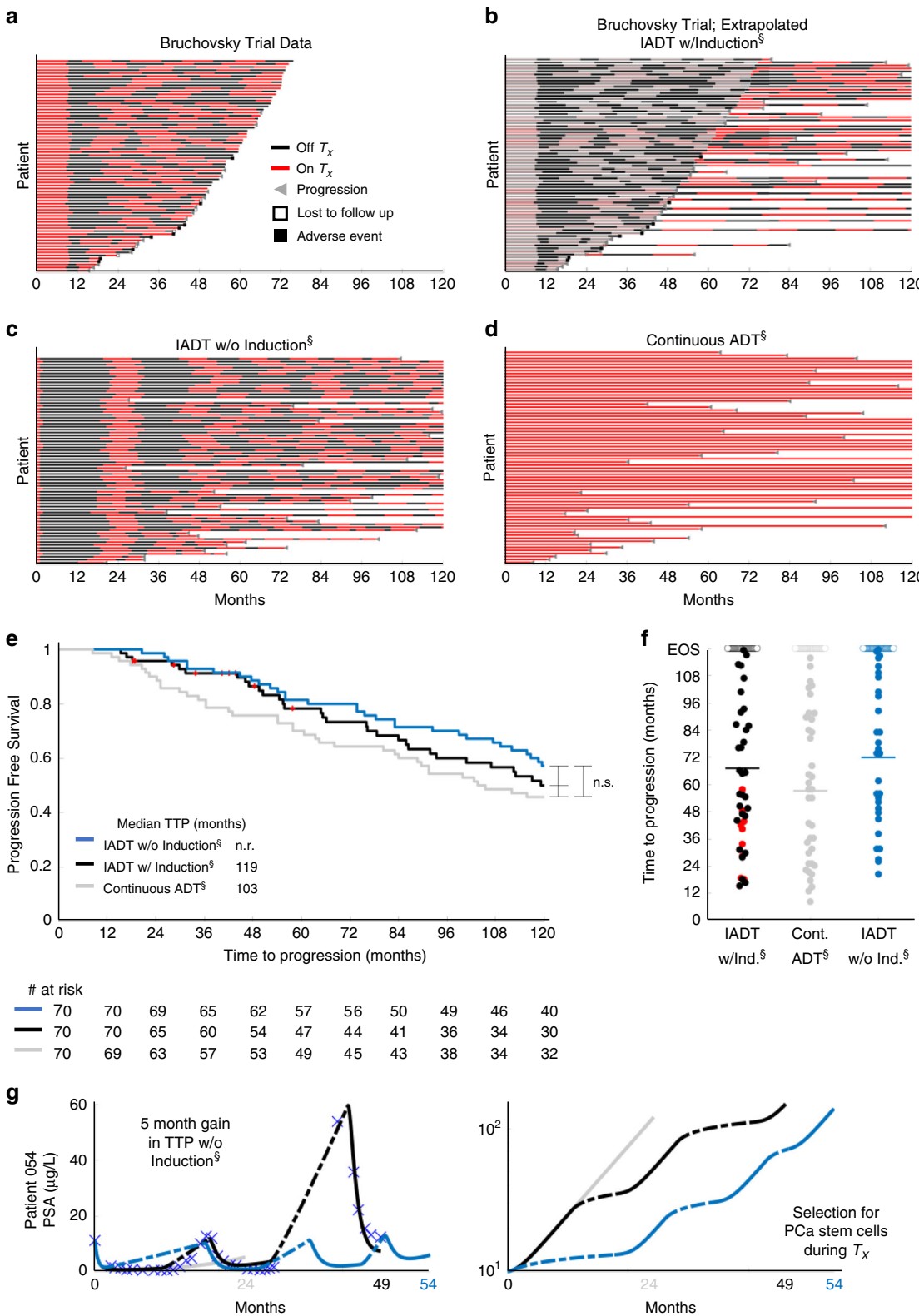

significant. The average TTP was ~4 months longer for IADT without the induction period when compared with the IADT with induction simulations (Fig. 5f) and could be increased by more than 5 months in select patients (Fig. 5g).

**Alternative treatment decision thresholds also increase TTP.** In the protocol used in the Bruchovsky study, treatment was paused after 36 weeks if PSA fell below 4 μg/L after both 24 and 32 weeks

of treatment. To investigate if a patient-specific approach that depends on individual pre-treatment PSA levels would be more effective, we simulated IADT using a 50% decline in PSA as a treatment cessation trigger, as well as 90% and 30% decline, and compared the results against those obtained in the Bruchovsky trial. Pausing treatment once PSA fell below 30% and 90% of the pre-treatment PSA, checking every two weeks, resulted in significantly longer TTP when compared with the IADT protocol

**Fig. 5 Administering IADT without induction period increases TTP when compared with IADT with induction and continuous ADT. a–d** Treatment times for **a** Bruchovsky trial data and simulations of **b** extrapolated IADT with induction, **c** IADT without induction, and **d** continuous ADT. Red and black denote when treatment is on and off, respectively. **e** Kaplan–Meier estimates of progression comparing IADT with induction (black curve) against IADT without induction (blue) and continuous ADT (gray) (double S indicates predictive model simulation). With and without induction period, IADT TTP increases when compared with continuous therapy TTP. Red symbols denote patients who experienced an adverse event/death prior to the conclusion of the trial. Statistical significance was determined using logrank test (IADT w/o induction vs. continuous: $p = 0.073$, IADT w/induction vs. continuous: $p = 0.129$, IADT w/induction vs. w/o induction: $p = 0.795$). **f** TTP comparison between IADT with induction (black), continuous therapy (gray), and IADT without induction (blue). Open circles denote end of simulation (EOS). Red dots denote patients who experienced an adverse event/death prior to the conclusion of the trial. Solid lines denote mean TTP (67.11, 57.25, and 71.87 months for IADT with induction, continuous therapy, and IADT without induction, respectively). The average TTP is longer with IADT without induction, compared with IADT with induction and continuous ADT. **g** Simulation results for Patient 054 (IADT with induction (black curve), without induction (blue curve), and continuous ADT (gray)). With IADT with induction, the patient became resistant after 49 months. Simulating continuous ADT would result in progression after 24 months. Simulating IADT without the induction would increase TTP by 5 months. PCaSC dynamics show that treatment selects for PCaSC population, accelerating resistance development. Dashed curve represent time when ADT is off.

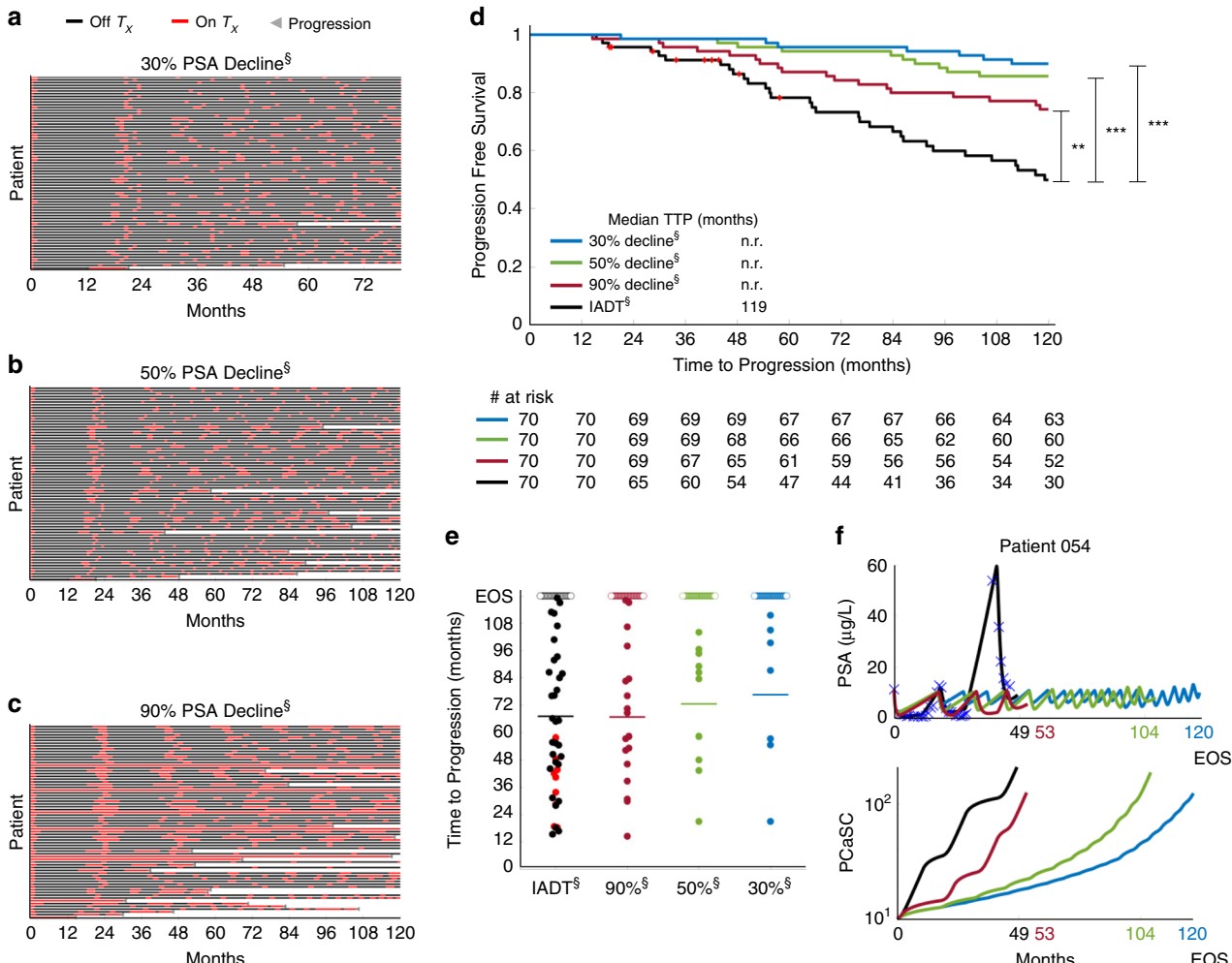

**Fig. 6 Alternative threshold therapy can significantly improve TTP. a–c** Treatment times for IADT with pre-treatment PSA decline triggers. Red and black denote when treatment is on and off, respectively. **d** Kaplan–Meier estimates of progression comparing standard IADT (black curve) with alternative therapy protocol using a 90% (maroon), 50% (green), and 30% (blue) PSA decline trigger (double S indicates predictive model simulation). All threshold levels show a significant increase in TTP when compared with standard IADT. Red symbols denote patients who experienced an adverse event/death prior to the conclusion of the trial. Statistical significance was determined using logrank test (IADT vs. 30% decline, 50% decline, and 90% decline: $p = 0.004$, $p = 5.154\text{E-}06$, $p = 7.929\text{E-}08$, respectively). Double and triple star denote $p < 0.01$ and $p < 0.001$, respectively. **e** TTP comparison between Bruchovsky IADT (black) and thresholds of 90%, 50%, and 30% PSA reduction. Open circles denote end of trial/simulation (EOS). Red dots denote patients who experienced an adverse event/death prior to the conclusion of the trial. Solid lines denote mean TTP (67.11, 66.79, 72.52, and 76.60 for standard IADT, 90%, 50%, and 30% PSA decline triggers, respectively). **f** Alternative threshold therapy simulations for Patient 054. With IADT, the patient became resistant after 49 months. Using a PSA decline of 90% and 50%, resistance could be delayed to 53 and 104 months, respectively. Using a PSA decline of 30%, the patient could continue on the protocol for >120 months.

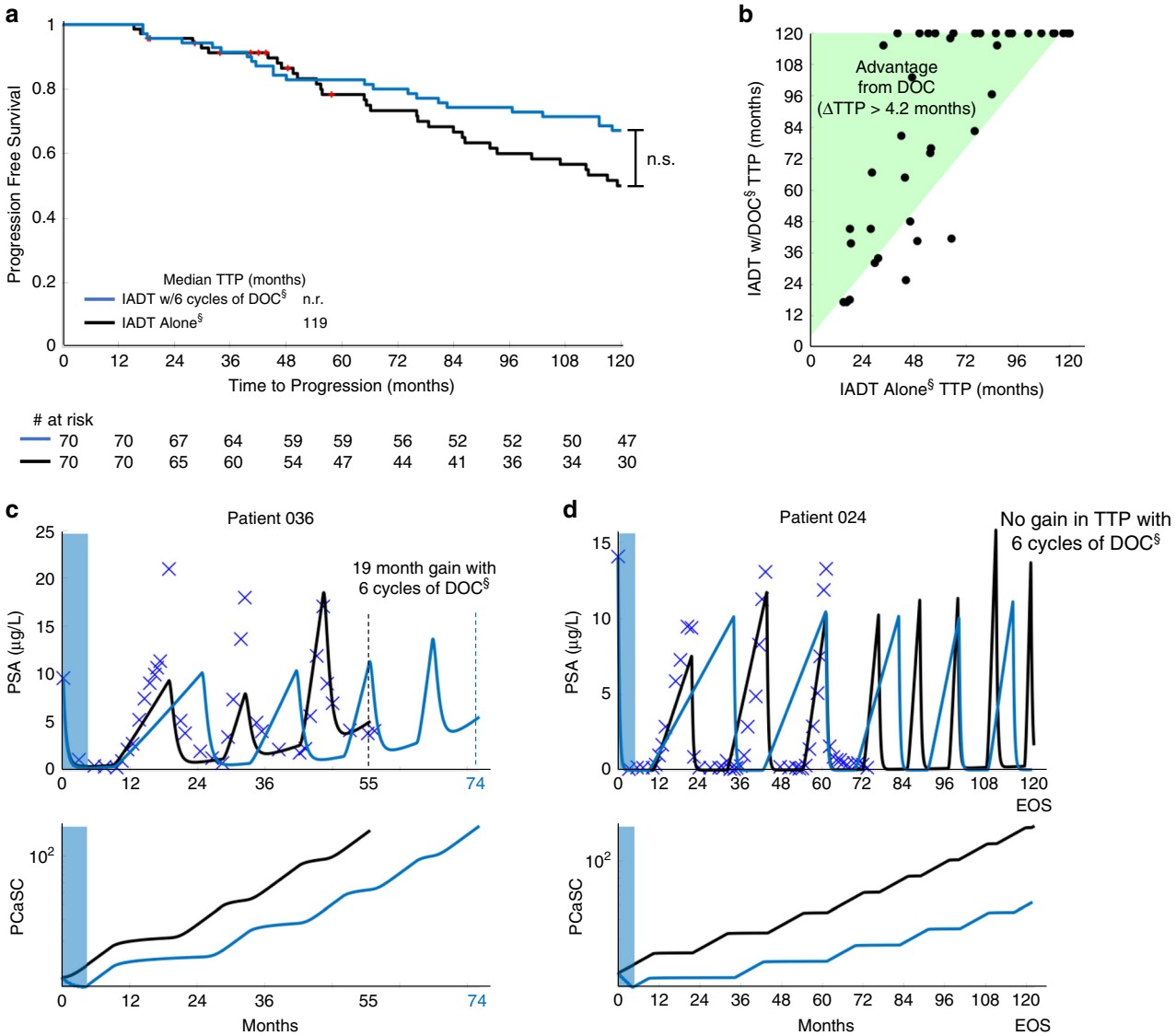

**Fig. 7 Docetaxel administration applied prior to IADT progression can increase TTP. a** Kaplan–Meier estimates of progression comparing IADT (black) against IADT with induction DOC (blue) (double S indicates predictive model simulation). Red symbols denote patients who experienced an adverse event/death prior to the conclusion of the trial. Statistical significance was determined using logrank test ($p = 0.260$). **b** TTP comparison between IADT alone and IADT with induction DOC. Shaded area shows where TTP for IADT with induction DOC is >4.2 months longer. **c–d** IADT simulations with (blue) and without (black) induction DOC. **c** With IADT alone, Patient 036 developed resistance after 55 months. IADT with induction DOC delayed progression for an additional 19 months. **d** Though the trial concluded after 6.5 years, simulations shows that Patient 024 could have remained on the IADT protocol for at least 10 years.

used in the trial (Fig. 6d–f). In addition, the average cumulative dose could be significantly reduced depending on the threshold used as shown in Fig. 6a–c. With a 90% decline from pre-treatment PSA, the average cumulative dose is ~64% of that of the standard IADT. A 50% decline results in ~30% of the cumulative dose used in standard IADT, whereas 30% results in ~27%. Controlling for the total time that each patient participated in the trial results in ~60%, 25%, and 22% of the cumulative dose used in standard IADT for thresholds of 90%, 50%, and 30%, respectively.

**Concurrent docetaxel administration provides favorable TTP.** As PCaSCs are resistant to ADT, it may be beneficial to administer alternative therapies that do not work directly via the AR pathway. AA, enzalutamide, apalutamide, and docetaxel (DOC) have shown benefit when used concurrently with ADT in various PCa settings, including both high- and low-volume disease in

metastatic hormone-sensitive PCa[27–31]. Thus, we sought to evaluate potential responses to combination therapy in the bio-chemically recurrent setting. We investigated the effect of con-current DOC by simulating six cycles of DOC early during IADT (as defined in the Methods section). Model analysis showed that administering DOC early with concurrent ADT followed by IADT may be able to increase TTP (Fig. 7a, c). In particular, model simulations showed that the majority of patients may receive a benefit of >4.2 months (length of six cycles of DOC) when given induction DOC with IADT (Fig. 7b). Equally important, simulations can identify selected patients for whom early concurrent DOC may not provide gains in TTP (Fig. 7d).

**First cycle PCaSC self-renewal rate stratifies patients who can benefit from DOC.** As shown above, the stem cell self-renewal rate plays a significant role in IADT and could accurately predict

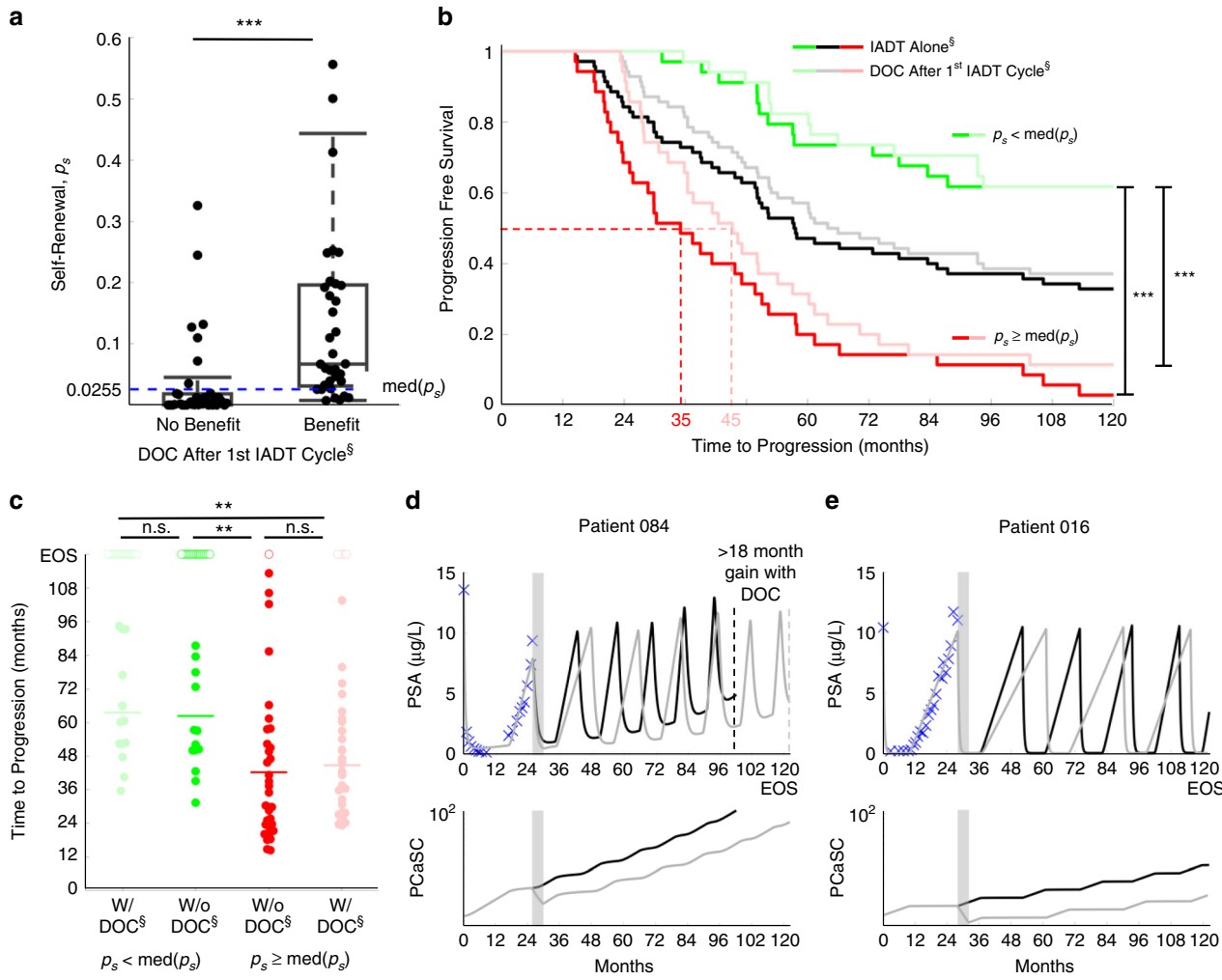

**Fig. 8 First cycle $p_s$ stratifies patients who could benefit from docetaxel after first cycle of IADT. a** $p_s$ distributions between patients who may benefit from docetaxel after first cycle of IADT and those who may not. Median $p_s$ (shown by blue dashed line) stratifies patients who could benefit from docetaxel after first cycle of IADT. The two-sample $t$ test was used to calculate the statistical significance of the difference between patients receiving no benefit ($n = 36$) and benefit ($n = 34$) from docetaxel after first cycle. Boxplots show median including the 25th and 75th percentiles. The two-sample $t$ test was used to calculate the statistical significance of the difference between the two groups ($p = 0.0004$). **b** Kaplan–Meier estimates of progression with and without docetaxel after first cycle of IADT (double S indicates predictive model simulation). Stratifying by median $p_s$ shows that patients with $p_s \geq \mathrm{med}(p_s)$ (red) experience greatest benefit from docetaxel (gain of 10 months). For patients with $p_s < \mathrm{med}(p_s)$ (green), there was not a significant benefit in TTP. Statistical significance was determined using logrank test. **c** TTP comparison between IADT with and without docetaxel. Open circles denote end of simulation (EOS). TTP was significantly shorter in patients with $p_s \geq \mathrm{med}(p_s)$ both with and without docetaxel. Solid lines denote mean TTP (63.57, 62.35, 42.26, and 44.77 months (left to right) for patients with low $p_s$ (green) and high $p_s$ (red), respectively). **d–e** Simulation results of IADT with (gray) and without (black) docetaxel after first cycle. **d** Patient 084 ($p_s = 0.0386 > \mathrm{med}(p_s)$) gains at least 18 months with docetaxel after first cycle. **e** Patient 016 ($p_s = 0.0017 < \mathrm{med}(p_s)$) did not progress with or without docetaxel. In **a–c**, double and triple star signifies $p < 0.01$ and $p < 0.001$, respectively.

a patient's response in subsequent cycles after just the first cycle. With this, we used the first cycle $p_s$ value to simulate six cycles of DOC with concurrent ADT followed by IADT and found that patients with $p_s \geq \mathrm{med}(p_s)$ could benefit from DOC after the first cycle of IADT (Fig. 8a). There was a significant difference in progression free survival between patients with a high $p_s$ and those with a low $\mathrm{p_s}$, both when simulating IADT with and without DOC after the first cycle (Fig. 8b). Though the difference in TTP was not significantly higher with DOC given after the first cycle (Fig. 8c), median TTP increased from 35 to 45 months in those patients with $p_s \geq \mathrm{med}(p_s)$ (Fig. 8b, d, e).

## Discussion
ADT is not curative for advanced PCa, as patients inevitably develop resistance. IADT is a promising approach to counteract evolutionary dynamics by reducing competitive release of the resistant subpopulation during treatment holidays. As IADT is highly dynamic, maximum efficacy requires continuous, accurate estimates of sensitive and resistant subpopulations.

Here, we present a simple mathematical model of evolutionary dynamics within biochemically recurrent PCa during IADT. The model has been trained with two parameters that are uniform across all patients and only two patient-specific parameters, which are interconnected, allowing us to further reduce them to a single, measurable parameter for each patient. Simulations suggest that, in this model, the evolution of PCaSCs is highly correlated with the development of resistance to IADT and may consequently be a plausible mechanism of resistance development. PSA dynamics suggest that resistant patients are likely to have higher PCaSC self-renewal rates than responsive patients, leading to increased production of PCaSCs and ultimately differentiated cells, thereby accelerating PSA dynamics with each

treatment cycle. Our results are similar to prior studies in glioblastoma that cancer stem cell self-renewal is likely to increase during prolonged treatment[32].

Using longitudinal PSA measurements and observed clinical outcomes from the IADT trial by Bruchovsky et al.[22], the model was calibrated to clinical data and predicted the development of resistance with 89% accuracy. Study analysis produced four important clinical findings: (1) PSA dynamics provide actionable triggers for PCa treatment personalization, vis-à-vis static PSA values with highly debated clinical utility[33]. (2) IADT outcomes in prior studies may be adversely affected by the long induction periods that accelerate selection for treatment-resistant subpopulations. (3) Patient-specific PSA treatment thresholds relative to pretreatment burden rather than a fixed value for all patients could significantly improve IADT responses. (4) Early treatment response dynamics during IADT can identify patients that may benefit from concurrent docetaxel treatment and, maybe more importantly, identify patients who are adequately treated with IADT alone.

Our study demonstrated the value of ongoing model simulations in predicting outcomes from each treatment cycle throughout the course of therapy. By integrating data that becomes available with each additional treatment cycle to adaptively inform tumor population dynamics, model simulations predict the response to the next cycle with a sensitivity and specificity of 73% and 91%, respectively and an overall accuracy of 89%. This ability to learn from early treatment cycles and predict subsequent responses adds an essential degree of personalization and flexibility to a cancer treatment protocol—a game theoretic strategy termed Bellman's Principle of Optimality that greatly increases the physician's advantage[34].

For those patients predicted to become resistant in the next cycle of IADT, an actionable model would also predict alternative treatments that could produce better clinical outcomes. The role of docetaxel in metastatic, hormone-sensitive PCa has been investigated in three studies in the past five years. The GETUG-AFU15 study found a non-significant 20% increase in overall survival in high volume disease (HVD) patients who received DOC concurrently with continuous ADT, but no survival benefit in those with low volume disease (LVD)[35]. Subsequently, the results of the STAMPEDE trial found that DOC administration resulted in a >12 months overall survival benefit with a median follow-up of 43 months[29]. Finally, the CHAARTED trial showed a statistically significant overall survival benefit from adding DOC in patients with HVD; however, no statistically significant survival benefit was found in LVD patients[30]. Here we explored the option of adding docetaxel in the treatment of biochemically recurrent PCa. Coinciding with previous findings, we found that concurrent docetaxel did not provide a significant benefit for the entire patient cohort. However, we found that analyzing PSA dynamics from the first IADT cycle stratifies patients who may receive benefit from concurrent administration of docetaxel. More importantly, the model was able to identify patients who would not receive a significant benefit in TTP from concurrent DOC treatment. These results emphasize that heterogeneity in patient responses can be harnessed by quantitative models to personalize treatment protocols.

Model validation showed that with two uniform parameters learned from our training cohort, only 63% of the data were accurately captured in the testing cohort. Although this may be perceived as a limitation of the model, the primary objective of this study was not to fit data with highest accuracy but to develop a predictive model that could be clinically actionable. Allowing for more patient-specific parameters substantially increases data fit, but compromises the ability to predict cycle-by-cycle dynamics as insufficient data are collected to inform each model parameter on a per-patient basis.

Since only 21% of the 70 patients progressed before the conclusion of the trial[22] (Supplementary Fig. 1), the clinical data included more responsive than resistant patients. This was overcome by bootstrapping the data, with more balanced data being poised to advance clinical utility. Of note, the presented model was calibrated and validated on patients with predominantly stage T2b and T3 disease, with Gleason scores ranging from 2 to 9[22]. As additional data become available, the developed framework may be generalizable and able to predict how PCa patients of different stages will respond to IADT with comparable accuracy. Further work in patients with more advanced or metastatic PCa is needed.

The mathematical model was developed with the intent to make clinically actionable predictions. To balance model complexity and prediction accuracy, the presented simple model has been derived[36]. Although numerous additional biological mechanisms are contributing to the observed PSA dynamics, it will become impossible to calibrate and validate their contributions from the measured clinical data. For reproducibility, we report the trained parameter values for the presented model. However, modification of the functional form of the developed model terms or inclusion of additional mechanisms will change each parameter. Therefore, herein estimated parameter values should not be taken as biological ground truth, nor are they translatable to different mathematical models.

A perceived limitation of the quantitative model may be the use of PSA dynamics as the sole biomarker of progression. PCa can become aggressive and metastatic despite low levels of serum PSA[37,38] with development of androgen independence, most notably in neuroendocrine PCa. Additional serum biomarkers such as circulating tumor cells (CTCs) and cell-free DNA (cfDNA) may prove useful in estimating intratumoral evolutionary dynamics in subsequent trials. With the CellSearch platform, higher CTCs enumeration more than five cells per 7.5 mL of peripheral blood has been shown to be prognostic and portend worse overall survival in metastatic CRPC patients[39]; however detecting CTCs in biochemically recurrent patients has been labor-intensive with low yield[40].

In conclusion, our study demonstrates that a simple mathematical model based on cellular dynamics in PCa can have a high predictive power in a retrospective data set from patients with biochemically recurrent PCa undergoing IADT. In particular, we demonstrate the model can use data from each treatment cycle to estimate intratumoral subpopulations and accurately predict the outcomes in subsequent cycles. Furthermore, in patients who are predicted to fail therapy in the next cycle, alternative treatments for which a response is more likely can be predicted. We conclude that PSA dynamics can prospectively predict treatment response to IADT, suggesting ways to adapt treatment to delay TTP.

## Methods

**IADT clinical trial data**. The Bruchovsky prospective Phase II study trial was conducted in 109 men with biochemically recurrent PCa[22]. IADT consisted of 4 weeks of Androcur as lead-in therapy, followed by a combination of Lupron and Androcur, for a total of 36 weeks. Treatment was paused if PSA had normalized (< 4 μg/L) at both 24 and 32 weeks, and resumed when PSA increased above 10 μg/L. PSA was measured every 4 weeks. Patients whose PSA had not normalized after both 24 and 32 weeks of being on treatment were classified as resistant and taken off of the study. We analyzed the data of 79 patients who had completed more than one IADT cycle. One patient was omitted for inconsistent treatment, seven were omitted owing to the development of metastasis and/or local progression, and one was omitted due to taking multiple medications throughout the trial, resulting in 70 patients included in the analysis (Supplementary Fig. 1). To calibrate and assess the accuracy of our model, the data was divided into training ($n = 35$, 27 responsive, eight resistant) and testing ($n = 35$, 28 responsive, seven resistant) cohorts, respectively matched for clinical response to treatment.

**Mathematical model of IADT response**. We developed a mathematical model of PCaSC (S), non-stem (differentiated) PCa cells (D), and serum PSA concentration

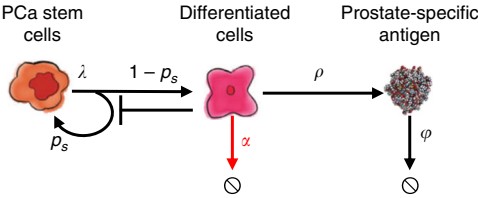

**Fig. 9 Model of prostate cancer stem-like cell and non-stem (differentiated) cell dynamics.** Interactions between stem-like and non-stem cells, as well as serum PSA. Stem-like cells can divide asymmetrically to produce differentiated cells. The differentiated cells inhibit the production of PCaSCs, die in response to ADT (shown by the red arrow), and produce PSA. Parameters are stem cell proliferation rate ($\lambda$), stem cell self-renewal $p_s$, ADT cytotoxicity $\alpha$, PSA production rate $\rho$, and PSA decay rate $\varphi$.

(P) (Fig. 9). PCaSCs divide with rate $\lambda$ (day$^{-1}$) to produce either a PCaSC and a non-stem PCa cell with probability $1-p_s$ (asymmetric division) or two PCaSCs at probability $p_s$ (symmetric division) with negative feedback from differentiated cells[16], modeled as $\frac{S}{S+D}$. Differentiated cells exclusively produce PSA at rate $\rho$ (μg/L day$^{-1}$), which decays at rate $\varphi$ (day$^{-1}$). Unlike androgen-independent PCaSCs, differentiated cells die in response to androgen removal at rate $\alpha$ (day$^{-1}$)[41]. IADT on and off cycles are described with parameter $T_x$, where $T_x = 1$ when IADT is given and $T_x = 0$ during treatment holidays.

The coupled mathematical equations describing these interactions are shown below

$$\frac{dS}{dt} = \left(\frac{S}{S+D}\right)p_s\lambda S, \quad \frac{dD}{dt} = \left(1 - \frac{S}{S+D}p_s\right)\lambda D - \alpha T_x D, \quad \frac{dP}{dt} = \rho D - \varphi P. \quad (1)$$

**Mathematical model training and validation.** Cell cycle duration is assumed to align with the evolved circadian rhythm[42]. Matsu-Ura et al. demonstrated that intercellular coupling helps stem cells to synchronize their cell divisions with local circadian pacemakers in secretory Paneth cells[43]. Thus, we set uninhibited PCaSCs to divide once per day at $\lambda = \ln(2)$. Sensitivity analysis[44] was done to assess the variation in the model output in response to small perturbations in the model parameters. This was computed by solving the sensitivity matrix $\chi = \frac{\partial P}{\partial \theta}$, where P is PSA and $\theta = (p_s, \alpha, \lambda, \varphi, \rho)$. Cell division rate $\lambda$ was the least sensitive parameter, whereas $\rho$ was the most sensitive (Supplementary Figure 3A–B). Correlation analysis[45] was used to determine parameter identifiability. Using the covariance matrix defined as $C = (\chi^T\chi)^{-1}$, the correlation coefficients were computed as $c_{ij} = \frac{C_{ij}}{\sqrt{C_{ii}C_{jj}}}$. If a parameter pair shows a strong correlation, that is if $\left|c_{ij}\right| > \xi$, then two parameters are correlated. In this case, we chose a correlation threshold of $\xi = 0.95$. Supplementary Figure 3C shows that $\lambda$ is highly correlated with $\rho$ and as the least sensitive parameter, we can confidently set $\lambda$ to be patient uniform at its nominal value of $\ln(2)$ and optimize the four remaining parameters (Supplementary Figure 3D).

We used particle swarming optimization (PSO)[46] to identify patient-specific model parameters that minimized the least squares error between model simulation and patient data. The parameter distributions obtained by calibrating the model to the data using these four parameters are shown in Supplementary Figure 4. Comparing the coefficients of variation between each parameter showed that $\varphi$ may not be required to be patient-specific and can be assumed uniform for all patients. Allowing the optimizer to find the uniform value of $\varphi$ and the remaining three patient-specific parameters produced accurate fits to the data (not shown). In addition, the coefficient of variation of $\rho$ revealed that this parameter could also be uniform between patients. Therefore, $\varphi$ and $\rho$ were set as uniform, whereas $p_s$ and $\alpha$ were patient-specific. PSO was used to identify population uniform and patient-specific model parameters in the training cohort. The trained mathematical model was assessed for accuracy in the validation cohort. The learned population uniform parameters were kept constant for all patients, and PSO was performed to find appropriate values for $p_s$ and $\alpha$ to produce accurate data fits. Resulting PCaSC proportion dynamics are shown in Supplementary Figure 5.

**Adaptive response prediction.** In order to predict the evolution of resistance, we started by fitting the model to each cycle of the training cohort data individually. That is, finding the optimal values of $p_s$ and $\alpha$, while allowing $\varphi$ and $\rho$ to remain fixed at the values previously found, to fit one cycle of data at a time (Fig. 2a). We then measured the relative change in $p_s$ between cycles and used this to generate cumulative probability distributions as shown in Fig. 2b. Sampling from the 95% confidence interval around the exponential curve relating $p_s$ to $\alpha$ in cycle $i+1$ (Fig. 2c), we found a corresponding $\alpha_{i+1}$. This $p_{s,i+1}$ and $\alpha_{i+1}$ were used to predict

the response in cycle $i+1$. In line with the trial by Bruchovsky et al.[22], resistance was defined as PSA increasing during treatment and/or a PSA level above 4 μg/L at both 24 and 32 weeks after the start of a cycle of treatment. Simulations that satisfied either of these conditions were classified as resistant. Thus, for each cycle prediction we obtained a probability of resistance ($P(\Omega)$ = proportion of resistance predictions). If the resulting $P(\Omega)$ was greater than the given threshold (obtained from the training cohort, Fig. 2d), then the prediction was classified as resistant. Otherwise, it was classified as responsive. The accuracy was computed as the proportion of predictions that were correctly classified.

**Modeling concurrent docetaxel.** Unlike ADT, docetaxel can induce cell death in both PCaSCs and non-stem cells, though to a lesser degree in PCaSCs compared with non-stem cells[47]. To model this, we extended the current model to include death of each cell type at rates $\beta_S$ (day$^{-1}$) and $\beta_D$ (day$^{-1}$). That is,

$$\frac{dS}{dt} = \left(\frac{S}{S+D}\right)p_s\lambda S - \beta_S T_{xD}S,$$

$$\frac{dD}{dt} = \left(1 - \frac{S}{S+D}p_s\right)\lambda D - \alpha T_x D - \beta_D T_{xD}S, \quad (2)$$

where $T_{xD} = 1$ when docetaxel is on and $T_{xD} = 0$ otherwise. Each cycle of docetaxel was simulated as a single dose on day $n$ ($T_{xD} = 1$) followed by 3 weeks without docetaxel ($T_{xD} = 0$). The parameters $\beta_S = 0.0027$ and $\beta_D = 0.008$ were chosen such that approximately three times more non-stem cells died than PCaSCs[47]. The data that support the findings of this study are available from the corresponding author upon reasonable request.

**Reporting summary.** Further information on research design is available in the Nature Research Reporting Summary linked to this article.

## Data availability
The clinical data used to conduct this study are available in a public repository at http://www.nicholasbruchovsky.com/clinicalResearch.html. A reporting summary for this article is available as a Supplementary Information file.

## Code availability
Code supporting the findings of this study are available at https://github.com/reneebrady/IADT_PCaSC.

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

## Acknowledgements

We thank participants of the clinical trial and Dr. Bruchovsky for sharing the data. We also thank our patient advocate Mr. Robert Butler for fruitful discussions. This work was supported by NIH/NCI 1R21CA234787-01A1 "Predicting patient-specific responses to personalize ADT for prostate cancer", and in part by NIH/NCI U54CA143970-05 (Physical Science Oncology Network) "Cancer as a complex adaptive system", the Ocala Royal Dames for Cancer Research, Inc., and The JAYNE KOSKINAS TED GIOVANIS FOUNDATION FOR HEALTH AND POLICY, a Maryland private foundation dedicated to effecting change in health care for the public good. The opinions, findings, and conclusions or recommendations expressed in this material are those of the authors and not necessarily those of the JAYNE KOSKINAS TED GIOVANIS FOUNDATION FOR HEALTH AND POLICY, its directors, officers, or staff.

## Author contributions

R.B., A.Z.W., T.Z., J.D.N., and H.E. conceptualized the study. R.B, J.D.N, T.A.G., and H.E. performed the modeling and statistical analyses. R.B., J.D.N., T.A.G., T.Z., A.Z.W., J.Z., R.A.G., and H.E. wrote the manuscript.

## Competing interests

Provisional patent application entitled "Methods for PCa intermittent adaptive therapy". Applicants/inventors: Renee Brady-Nicholls, Heiko Enderling; application number: 62/944.804; status: provisionally filed. The patent covers methods related to using the mathematical model to adjust an individual patient's treatment administration (both timing and alternative treatment options), thereby increasing TTP. T.Z.: Research funding: Acerta, Novartis, Merrimack, Abbvie/StemCentrx, Merck, Regeneron, Mirati Therapeutics, Janssen, Astra Zeneca, Pfizer, OmniSeq, Personal Genome Diagnostics, Seattle Genetics; Speakers Bureau: Genentech/Roche, Exelixis, Sanofi-Aventis, Genomic Health; Advisory Board: Genentech/Roche, Merck, Exelixis, Sanofi-Aaventis, Janssen, Astra Zeneca, Pfizer, Amgen, BMS, Pharmacyclics, Seattle Genetics, Bayer; Consultant: Bayer, Astra Zeneca, Foundation Medicine; Employee: Capio Biosciences, Archimmune Therapeutics (spouse); Stockholder: Capio Biosciences, Archimmune Therapeutics (spouse); A.Z.W.: Cofounder, stockholder and consultant: Capio Biosciences, Archimmune Therapeutics; J.Z.: Consultant: Dendreon, Advisory Board: AstraZeneca, Bayer, Clovis Oncology, Seattle Genetics, Speaker Bureau: Merck, Sanofi. All other authors declare no competing interests.
