## [Peer Review File · Nature Communications]

Reviewers' comments:

Reviewer #1 (Remarks to the Author); expert in mathematical modelling:

Reviewer Comments for Prostate-Specific Antigen Dynamics Predict Individual Responses to Intermittent Androgen Deprivation

This paper puts forth a new ODE model for prostate cancer stem cell dynamics during treatment, which the authors use to infer important parameters including PCaSC self-renewal bias and ADT cytotoxicity from longitudinal PSA dynamics for each patient in the Bruchovsky 2006 IADT trial. The central hypothesis, that PCaSCs become enriched in patient treated with ADT and contribute to treatment failure, is timely and important. This hypothesis is not tested nor confirmed (as the paper does not include clinically measured PCaSC fractions or growth rates), but it is translated into a mathematical model that allows the authors to simulate and evaluate several clinically relevant variations of IADT treatment design. Overall, the paper is well-written and will be of interest to researchers in the field.

Major concerns:

1. If possible, the authors should use bootstrapping to split the training and testing sets. This would reduce possible sampling bias – currently the inferred α values are quite different between the testing set than the training set (figure S2 vs. figure 1). This could also allow the authors to evaluate the robustness of their findings, including the estimates for the PSA production and decay rates.
2. The authors report best fit parameter values but do not report confidence intervals on these estimates. There is also currently no discussion as to whether the best fit parameter values are biologically reasonable. In particular, the best fit values for the probability of symmetric self-renewal p_S vary between $2.2E-14$ and 0.26 , which seems unrealistically wide a range, and the cytotoxicity rate has to a lesser extent the same issue, varying between 0.0067 and 1 .
3. The authors infer PCaSC levels over time for each patient but cannot verify these levels, so it is reaching to say that the model simulations support that idea that PCaSCs are involved in TTP. Rather, it seems to me that the simulations assume that PCaSCs are involved, and given this assumption, can predict patient outcomes and compare treatment designs. If the authors agree, the second paragraph of the discussion should be rephrased to reflect this, especially the statement that “Model simulations support the central hypothesis that the evolution of PCaSCs is highly correlated with the development of resistance to IADT.” It would help to also clarify that “Resistant patients are likely to have higher inferred PCaSC self-renewal rates.”
4. The ODE model does not account for the reproduction of differentiated cells nor the natural death of stem cell and differentiated cell populations. Either these assumptions should be better justified, or the model should be generalized to include these rates. By allowing only PCaSCs to reproduce in the model, it seems somewhat trivial that resistant patients are found to also have higher inferred levels of PCaSCs, as this correlation may simply reflect the model assumptions.
5. Were this prediction approach to be applied in practice, the threshold values κ likely would be of particular importance, but these values are seem to be given only in Figure 8 and not discussed in the main text. In particular, it is unclear why the optimal κ_2 is much lower than the optimal κ_1 and κ_3 . This may become clearer if bootstrapping were applied.
6. The simulation results rely on the parameter choice that PCaSCs divide once per day, for which the authors cite reference 22, but this reference is not specific to PCaSCs, or even cancer cells, or even human cells. A reference more relevant to PCaSCs, or at least cancer stem cells, will be crucial, and this parameter choice may need to be updated.

Minor concerns:

1. The paper relies on the claim that PCaSCs are less sensitive to ADT than PCaCs, such that their expansion during treatment can accelerate TTP. The model explores the consequence of this claim but does not provide direct support for it. It would benefit the paper to spend more than one sentence in the introduction (bottom of page 3) describing the findings of references 18-20, which directly support this central claim.

2. The simplicity of the ODE system, with relatively few parameters, is a strength of the paper and should be emphasized. In particular, it should be noted that the fraction of PCaSCs, $s=S/(S+D)$, satisfies the ODE $ds/dt = \alpha Txs(1-s) - (1-pS)\lambda s^2$, which converges to a nonzero steady state value during treatment and converges to 0% between cycles. In the methods section, the authors should clarify that the negative feedback from differentiated cells changes the fraction of symmetric divisions (rather than the division rate itself); pS should not be called the "rate" of symmetric division but rather pS times $S/(S+D)$ should be called the probability of symmetric division. If the negative feedback was of even slightly lower order, such that $dS/dt = \alpha n p S \lambda S$ where $n < 1$, then the PCaSC fraction s would converge to a nonzero value even in the absence of treatment. It would be beneficial to explain why $n=1$ was chosen.

3. The authors should provide some justification for why they assume that PSA production and decay rates do not vary between patients, or else run a few simulations in which they allow these rates to vary between patients to confirm that they fall within a narrow range of values.

4. The paper shows a correlation between the inferred values of pS and α but does not report any statistics about this correlation (such as an R^2 value) nor justify why an exponential model was chosen. The statement "This relationship allows prediction of future cycles based on PCaSC self-renewal rate pS as the single, identifiable, independent patient-specific parameter" suggests a high correlation, but then is at odds with the claim that treatment resistance is caused by a high pS value, since it might instead be caused by a low α value.

5. Figure 2 shows that the model can be used to predict whether an individual patient should be advised to stop the trial, and figure 4 shows that the model can be used to choose a PSA threshold level for IADT, but ideally one would choose a different threshold level for each patient after the first cycle, and this does not seem to be simulated despite the statement in the discussion that "(2) With IADT, applying a PSA treatment threshold that depends on pre-treatment PSA levels (rather than a fixed value for all patients) can significantly increase TTP."

6. Most of the figures include a plot of the PCaSC level S over time, but it would be preferable to replace each of these plots with a plot of the PCaSC fraction s over time. The legends of figures 1 and S2 should state that gray shading indicates the treatment window. The color used for resistance is red in figures 1 and S2 but blue in figure 5, which may be confusing. In figure 3A-B, the red dots do not seem to be described in the legend, and the left plot of Fig 3C should make use of dashed lines to correspond to right plot. In figure 8B, the horizontal axis would benefit from being on a log scale.

Reviewer #2 (Remarks to the Author); expert in prostate therapy:

General comments:

The authors present data about how prostate specific antigen (PSA) dynamics predicts individual response to intermittent androgen deprivation (IADT). They developed the model based on phase II study in 109 men with biochemically recurrence after radiotherapy for locally advanced prostate

cancer. In the end, 70 patients were included in the analysis and divided into a training set (n= 27 responsive, 8 resistant patients) and validation cohort (28 responsive, 7 resistant patients). The mathematical model of PCaSC proliferation was based on the PSA dynamics without other objectives parameters. While this is an extremely interesting concept and has merit, unfortunately the model is only as good as the data set you develop it from. This is a very homogenous data set of patients that were treated in a selected phase II study and the model was based on only 8 resistant patients that failed treatment. In addition, the concern of this data base is many of these patients may have received prior neo-adjuvant ADT with radiotherapy (neo-adjuvant androgen-suppression therapy ≤ 3 months' duration was allowed) which does complicate this set of patients further. So the challenge is how this model will perform in a larger more contemporary\heterogeneous data set in other men with prostate cancer and can this be generalizable. If not, then the model has minimal utility. I do applaud the authors for applying mathematical models to prostate cancer data sets.

Specific comments:

1. Abstract: To infer that IADT will delay development of treatment resistance has only been shown in non-prostate cancer model and not in human trials. Please re-phrase.
2. Page 5, paragraph 1: PCaSC divide approximately once per day- please provide additional direct reference. Please also clarify how did you estimate all these variables and how do you compensate in the model for various stage, grade, histology, prior therapy or other factors which can influence PSA secretion (ie testosterone level at baseline). Much of the model is based on the PCaSC renewal rate based on the PSA dynamics. Since the PSA is AR gene driven, how do you account for "low PSA secreting cells" in your model and the presence of AR negative phenotype in the tumor which may not reflected in the serum PSA?
3. Page 5, second paragraph: Your model had an R2 value of .69. This means 30% of the results are not accountable. If you tested this in a larger more heterogeneous data set would this still be predictive?
4. Page 6, Paragraph 1: The model yielded a sensitivity of 57% and specificity of 94% with an accuracy of 90%. Please explain why with such a low sensitivity (which most models we would expect a high sensitivity not low) the model can give an accuracy of 90%.
5. Page 8: Ongoing pilot study (NCT02415621): It is not appropriate to analyze and published an ongoing trial since this may bias the clinical study. Would remove this section since this also adds little to the publication. The following paragraph in this section is also irrelevant to the manuscript and does not pertain to contemporary patients.
6. Page 9, Concurrent docetaxel administration provides favorable TTP: As the authors states many trials have defined populations which concomitant docetaxel therapy with ADT have shown clinical benefit with several more in early settings ongoing. Not sure how this adds to current knowledge that exists.

We would like to thank both reviewers for critical reading of our manuscript. We greatly appreciate all of the reviewers' insightful comments and suggestions, which helped tremendously to improve the quality of the manuscript. Please find our point-by-point response to each of the reviewers' concerns below.

Reviewers' comments:

Reviewer #1 (Remarks to the Author); expert in mathematical modelling:

Reviewer Comments for Prostate-Specific Antigen Dynamics Predict Individual Responses to Intermittent Androgen Deprivation

This paper puts forth a new ODE model for prostate cancer stem cell dynamics during treatment, which the authors use to infer important parameters including PCaSC self-renewal bias and ADT cytotoxicity from longitudinal PSA dynamics for each patient in the Bruchovsky 2006 IADT trial. The central hypothesis, that PCaSCs become enriched in patient treated with ADT and contribute to treatment failure, is timely and important. This hypothesis is not tested nor confirmed (as the paper does not include clinically measured PCaSC fractions or growth rates), but it is translated into a mathematical model that allows the authors to simulate and evaluate several clinically relevant variations of IADT treatment design. Overall, the paper is well-written and will be of interest to researchers in the field.

Major concerns:

1. If possible, the authors should use bootstrapping to split the training and testing sets. This would reduce possible sampling bias “ currently the inferred $\hat{\pm}$ values are quite different between the testing set than the training set (figure S2 vs. figure 1). This could also allow the authors to evaluate the robustness of their findings, including the estimates for the PSA production and decay rates.
Response: *We agree with the reviewer that bootstrapping is an effective tool to estimate statistics of a particular dataset. It does, however, require sampling with replacement, which may result in the model being tested on data that it was previously trained on. We chose to use stratified random sampling to split the training and testing cohorts instead as this does not require any resampling at the risk of testing on a patient that the model was previously trained on. We have revised the manuscript to include the stratified random sampling details and included a reference of this particular method (page five, first paragraph).*
2. The authors report best fit parameter values but do not report confidence intervals on these estimates. There is also currently no discussion as to whether the best fit parameter values are biologically reasonable. In particular, the best fit values for the probability of symmetric self-renewal p_S vary between $2.2E-14$ and 0.26 , which seems unrealistically wide a range, and the cytotoxicity rate has to a lesser extent the same issue, varying between 0.0067 and 1 .
Response: *Thank you for bringing this to our attention. The parameter confidence intervals have been added (page five, first paragraph). In regards to the values of the parameters, we are not able to test the biological relevance of the parameters due to their actual values not being available in literature. As such, we are cautioned to say that this is the ground truth for a particular parameter since adding or removing components from the model would likely result in different parameters. From the design of the model, we can say that those patients with a low p_S value are likely evolutionarily stable. Similarly, patients with a high α value ($\alpha \rightarrow 1$) are likely responding very well*

to treatment, regardless of their stem cell dynamics. We have discussed this in the revised manuscript (page six, first paragraph).

3. The authors infer PCaSC levels over time for each patient but cannot verify these levels, so it is reaching to say that the model simulations support that idea that PCaSCs are involved in TTP. Rather, it seems to me that the simulations assume that PCaSCs are involved, and given this assumption, can predict patient outcomes and compare treatment designs. If the authors agree, the second paragraph of the discussion should be rephrased to reflect this, especially the statement that “Model simulations support the central hypothesis that the evolution of PCaSCs is highly correlated with the development of resistance to IADT.” It would help to also clarify that “Resistant patients are likely to have higher inferred PCaSC self-renewal rates.”

Response: We thank the reviewer for this correct observation. The manuscript has been revised to reflect this (page 11, first paragraph). The text now reads “ Simulations suggest that, in this model, the evolution of PCaSCs is highly correlated with the development of resistance to IADT and may consequently be a plausible mechanism of resistance development.

4. The ODE model does not account for the reproduction of differentiated cells nor the natural death of stem cell and differentiated cell populations. Either these assumptions should be better justified, or the model should be generalized to include these rates. By allowing only PCaSCs to reproduce in the model, it seems somewhat trivial that resistant patients are found to also have higher inferred levels of PCaSCs, as this correlation may simply reflect the model assumptions.

Response: This is correct. However, our primary objective was to develop the simplest model possible that could be used to make accurate predictions of subsequent responses to treatment. We have included a discussion to this extent in the revised manuscript. The model results lend support to our hypothesis that stem cell self-renewal dynamics could be a plausible mechanism that is driving resistance.

5. Were this prediction approach to be applied in practice, the threshold values $\hat{\rho}$ likely would be of particular importance, but these values are seem to be given only in Figure 8 and not discussed in the main text. In particular, it is unclear why the optimal $\hat{\rho}_2$ is much lower than the optimal $\hat{\rho}_1$ and $\hat{\rho}_3$. This may become clearer if bootstrapping were applied.

Response: The reviewer is right that the values of the prediction thresholds are very important. We have described the methods used to determine thresholds for each cycle in detail in the Materials and Methods section. The optimal κ_2 is lower than the other thresholds due to very few patients progressing during their second cycle of treatment. This resulted in very low probabilities of resistance for the majority of the patients, forcing κ_2 to be relatively small compared to the κ_3 and κ_4 where more patients developed resistance per cycle. We have also discussed this in the revised manuscript discussion.

6. The simulation results rely on the parameter choice that PCaSCs divide once per day, for which the authors cite reference 22, but this reference is not specific to PCaSCs, or even cancer cells, or even human cells. A reference more relevant to PCaSCs, or at least cancer stem cells, will be crucial, and this parameter choice may need to be updated.

Response: We agree that support for this assumption is necessary. Cell-cycle duration is assumed to align with the evolved circadian rhythm (Glass, Nature, 2001). A study by Matsu-Ura et al. showed that stem cells synchronize their cell divisions with circadian rhythms via pacemakers in the secretory Paneth cells. This coupling was shown to be 1:1 or 1:2 depending on the cell subpopulation. We have added these references and discussion in the revised manuscript (page 16, last paragraph).

Additionally, we used sensitivity and correlation analysis to show that this parameter is the least sensitive parameter and is highly correlated with the most sensitive parameter. Therefore, we chose to set this parameter at its nominal values and optimize the remaining parameters. We have included Figs. S2 and S3 showing the results of these analyses, as well as a description of this analysis in the manuscript (page 17, first paragraph).

Minor concerns:

1. The paper relies on the claim that PCaSCs are less sensitive to ADT than PCaCs, such that their expansion during treatment can accelerate TTP. The model explores the consequence of this claim but does not provide direct support for it. It would benefit the paper to spend more than one sentence in the introduction (bottom of page 3) describing the findings of references 18-20, which directly support this central claim.

Response: *The introduction has been revised to describe the findings of reference 18-20. The updated text now reads "A pre-clinical study by Bruchovsky et al. showed ADT selects for murine PCaSCs (19). Analogously, Lee et al. demonstrated increased PCaSCs populations after ADT in patient-derived PCa cell lines, which can be reverted by the addition of functional AR (20). Combined, these results suggest evolution of or selection for pre-existing androgen-independent PCaSCs as a plausible explanation of the development of ADT resistance.*

2. The simplicity of the ODE system, with relatively few parameters, is a strength of the paper and should be emphasized. In particular, it should be noted that the fraction of PCaSCs, $s = S/(S+D)$, satisfies the ODE $ds/dt = \hat{\lambda} \pm Txs(1-s) - (1-pS)\hat{\lambda}s^2$, which converges to a nonzero steady state value during treatment and converges to 0% between cycles. In the methods section, the authors should clarify that the negative feedback from differentiated cells changes the fraction of symmetric divisions (rather than the division rate itself); pS should not be called the "rate" of symmetric division but rather pS times $S/(S+D)$ should be called the probability of symmetric division. If the negative feedback was of even slightly lower order, such that $dS/dt = \text{sn}pS\hat{\lambda}S$ where $n < 1$, then the PCaSC fraction s would converge to a nonzero value even in the absence of treatment. It would be beneficial to explain why $n=1$ was chosen.

Response: *This is correct. We have changed the wording in the text to reflect that the probability of symmetric division depends on the stem cell fraction (page 16, second paragraph). We have also added Fig. S5 showing that the stem cell fraction converges to a nonzero steady state during treatment and to zero when treatment is turned off. We have also added an additional discussion about model simplicity in the context of reported parameter values as mentioned above (page 13, final paragraph).*

3. The authors should provide some justification for why they assume that PSA production and decay rates do not vary between patients, or else run a few simulations in which they allow these rates to vary between patients to confirm that they fall within a narrow range of values.

Response: *We have added Fig. S3 to show the spread of the parameter values when all four are allowed to be patient-specific. The coefficient of variation showed that ϕ and ρ had the least variation among all patients and was therefore assumed to be uniform between all patients. The manuscript has been revised to include this analysis (page 17, first paragraph).*

4. The paper shows a correlation between the inferred values of pS and $\hat{\lambda}$ but does not report any statistics about this correlation (such as an R^2 value) nor justify why an exponential model was chosen. The statement "This relationship allows prediction of future cycles based on PCaSC self-renewal rate pS as the single, identifiable, independent patient-specific parameter" suggests a

high correlation, but then is at odds with the claim that treatment resistance is caused by a high p_s value, since it might instead be caused by a low $\hat{\mu}$ value.

Response: A supplementary table has been added to show the comparison between various alternative functional forms using the mean squared error. Though both p_s and α are both patient specific parameters, analysis of the training set showed that α was not significantly different between the responsive and resistant patients (Fig. S4D). As such, we must focus on p_s as the single, identifiable, patient-specific parameter and use the exponential relationship to find the corresponding value for α .

5. Figure 2 shows that the model can be used to predict whether an individual patient should be advised to stop the trial, and figure 4 shows that the model can be used to choose a PSA threshold level for IADT, but ideally one would choose a different threshold level for each patient after the first cycle, and this does not seem to be simulated despite the statement in the discussion that “(2) With IADT, applying a PSA treatment threshold that depends on pre-treatment PSA levels (rather than a fixed value for all patients) can significantly increase TTP.”

Response: In Fig. 2, we are showing that cycle by cycle dynamics can be used to determine whether a particular patient would continue to respond to IADT under the protocol used in the study. That is, treatment is paused once PSA falls below a threshold of 4 $\mu\text{g/L}$ at both 24 and 32 weeks. In Fig. 4, we are simulating using a patient-specific treatment decision trigger that is based on an individual patient’s pre-treatment PSA level, rather than a set value of 4 $\mu\text{g/L}$. The text has been revised to clarify this point (page nine, first paragraph).

6. Most of the figures include a plot of the PCaSC level S over time, but it would be preferable to replace each of these plots with a plot of the PCaSC fraction s over time. The legends of figures 1 and S2 should state that gray shading indicates the treatment window. The color used for resistance is red in figures 1 and S2 but blue in figure 5, which may be confusing. In figure 3A-B, the red dots do not seem to be described in the legend, and the left plot of Fig 3C should make use of dashed lines to correspond to right plot. In figure 8B, the horizontal axis would benefit from being on a log scale.

Response: We have included Fig. S5 to show the change in the PCaSC fraction over time for each of the patients shown in Figs. 1 and S4 (formerly S2). The legends of Figs. 1 and S4 have been revised. Figs. 3 and 5 have also been updated. Fig. 8B has been revised to more clearly show the scale.

Reviewer #2 (Remarks to the Author); expert in prostate therapy:

General comments:

The authors present data about how prostate specific antigen (PSA) dynamics predicts individual response to intermittent androgen deprivation (IADT). They developed the model based on phase II study in 109 men with biochemical recurrence after radiotherapy for locally advanced prostate cancer. In the end, 70 patients were included in the analysis and divided into a training set ($n=27$ responsive, 8 resistant patients) and validation cohort (28 responsive, 7 resistant patients). The mathematical model of PCaSC proliferation was based on the PSA dynamics without other objective parameters. While this is an extremely interesting concept and has merit, unfortunately the model is only as good as the data set you develop it from. This is a very homogenous data set of patients that were treated in a selected phase II study and the model was based on only 8 resistant patients that failed treatment. In addition, the concern of this data base is many of these patients may have received prior neo-adjuvant ADT with radiotherapy (neo-adjuvant androgen suppression therapy

3 months' duration was allowed) which does complicate this set of patients further. So the challenge is how this model will perform in a larger more contemporary\heterogeneous data set in other men with prostate cancer and can this be generalizable. If not, then the model has minimal utility. I do applaud the authors for applying mathematical models to prostate cancer data sets.

Response: *We thank the reviewer for raising this valid concern. Unfortunately, we can only build a model based on the data that is currently available to us. The model is calibrated and validated to data from patients with predominantly stage T2B and T3 disease. Gleason scores also ranged between 2 and 9 (52.4% ≤ 6, 47.6% >6). As more data becomes available, we are confident that we will be able to generalize the model to be able to predict how PCa patients of different stages will respond to IADT with comparable accuracy. The discussion has been revised to include this point (page 13, second paragraph). The updated text now reads "Of note, the presented model was calibrated and validated on patients with predominantly stage T2b and T3 disease, with Gleason scores ranging from 2-9 (21). As additional data become available, the developed framework may be generalizable and able to predict how PCa patients of different stages will respond to IADT with comparable accuracy."*

Specific comments:

1. Abstract: To infer that IADT will delay development of treatment resistance has only been shown in non-prostate cancer model and not in human trials. Please re-phrase.

Response: *Thank you. The abstract has been revised to reflect that this has only been shown in mice. Thank you for bringing it to our attention.*

2. Page 5, paragraph 1: PCaSC divide approximately once per day- please provide additional direct reference. Please also clarify how did you estimate all these variables and how do you compensate in the model for various stage, grade, histology, prior therapy or other factors which can influence PSA secretion (ie testosterone level at baseline). Much of the model is based on the PCaSC renewal rate based on the PSA dynamics. Since the PSA is AR gene driven, how do you account for "low PSA secreting cells" in your model and the presence of AR negative phenotype in the tumor which may not be reflected in the serum PSA?

Response: *We have updated the manuscript to include direct references for our assumption that PCaSCs divide approximately once per day (page 16, last paragraph). Additionally, we used sensitivity and correlation analysis to show that this parameter is the least sensitive parameter and is highly correlated with the most sensitive parameter. Therefore, we chose to set this parameter at its nominal values and optimize the remaining parameters. We have a dedicated section in the revised manuscript to discuss parameter estimation (page 17, first paragraph), and included Figs. S2 and S3 showing the results of these analyses. We also added an additional discussion that the absolute parameter values are always an artifact of the chosen model. We do not aim to report a translatable parameter value; rather we report the values for the presented model that yields the prediction accuracy based on measurable PSA dynamics. In our model, PCaSC are considered to be non- PSA secreting cells. Therefore, an AR negative phenotype in the tumor is accounted for by a higher PCaSC population.*

3. Page 5, second paragraph: Your model had an R2 value of .69. This means 30% of the results are not accountable. If you tested this in a larger more heterogeneous data set would this still be predictive?

Response: *The primary objective of our study was not to fit data with the highest accuracy, but to develop a predictive model that could be clinically actionable. Model analysis shows that allowing for more patient-specific parameters will substantially improve the model's fit to the data, however this*

compromises the ability to predict individual responses to subsequent cycles of treatment. We have extended the discussion in the revised manuscript to reflect that (page 14, second paragraph).

4. Page 6, Paragraph 1: The model yielded a sensitivity of 57% and specificity of 94% with an accuracy of 90%. Please explain why with such a low sensitivity (which most models we would expect a high sensitivity not low) the model can give an accuracy of 90%.

Response: *In our study, sensitivity referred to correctly predicting resistance, while specificity referred to predicting response. As there were more responsive patients than resistant patients in each cycle, we would expect that the specificity would have more weight on the accuracy than the sensitivity. This discussion has been added to the revised manuscript (page 13, second paragraph) as "Since only 21% of the 70 patients progressed before the conclusion of the trial (21) (Fig. S1), the clinical data included more responsive than resistant patients in both the training and testing cohorts. Consequently, the model is able to predict response more accurately than resistance, as demonstrated by the relatively low sensitivity at 57% compared to a specificity of 94%. Combined, these result in a high overall accuracy."*

5. Page 8: Ongoing pilot study (NCT02415621): It is not appropriate to analyze and published an ongoing trial since this may bias the clinical study. Would remove this section since this also adds little to the publication. The following paragraph in this section is also irrelevant to the manuscript and does not pertain to contemporary patients.

Response: *Thank you for bringing this to our attention. The ongoing trial results have been removed.*

6. Page 9, Concurrent docetaxel administration provides favorable TTP: As the authors states many trials have defined populations which concomitant docetaxel therapy with ADT have shown clinical benefit with several more in early settings ongoing. Not sure how this adds to current knowledge that exists.

Response: *While we agree that concomitant docetaxel therapy with ADT has shown clinical benefit in prior studies, few have identified markers to determine who may and may not benefit from such therapy. Using our model of PSA dynamics obtained from just one cycle of IADT, we are able to identify who may and more importantly, who may not benefit from concurrent DOC. As chemotherapy can be quite toxic, it is important to identify those who can be spared such therapy. We have included this discussion in the revised manuscript to emphasize the importance of these results (page 12, second paragraph).*

Reviewers' comments:

Reviewer #1 (Remarks to the Author):

Thank you to the authors for taking the time to address many of our initial concerns. The paper has been improved during revision, and a few of our concerns still remain as outlined below.

Major concerns:

1. An open question remains to what extent this paper's conclusions might change if the random assignment of patients to the training and testing groups had been different. If possible, could the authors either (a) split the patients again randomly to obtain new training and testing sets or (b) switch the patients in the current training and testing sets, in order to ensure that the results are not sensitive to this random assignment?

2. Because the authors were unable to find parameter values from the literature, it would be useful to include values for at least the self-renewal probability of other types of stem cells, which have been inferred by several papers directly from division data. Also, it seems that the 95% intervals reported are interquartile ranges rather than confidence intervals. These are also useful, but I think it would be even more important to report the confidence interval for each inferred parameter. For example, with 95% confidence can you conclude that the PSA decay rate falls into some interval that you can report? Currently only the best estimate $\phi = 0.0856$ is reported. Likewise, can you report a confidence interval for the PCaSC self-renewal rate of patient 091, for example? Currently only the best estimate $pS = 0.0201$ is reported; a confidence interval would help address whether patient heterogeneity in pS and α is due to inference uncertainty or real patient differences.

3. This concern has been fully addressed.

4. A reader might be concerned that the ODE model would behave quite differently if cell death or the reproduction of differentiated cells were to be included. However, perhaps you can provide the following argument for why they can both be neglected: If stem cells die with rate d_1 , and if differentiated cells die with rate d_2 and reproduce with rate r , then the dS/dt equation would include an additional $-d_1 \cdot S$ term, and the dD/dt equation would include an additional $+(r-d_2)D$ term. But then the ODE describing the fraction of stem cells $s=S/(S+D)$ would be $ds/dt = (aT_x - r - d_1 + d_2)s(1-s) - (1-p) \cdot \lambda \cdot s^2$. If you assume that $r = d_1 - d_2$ during treatment holidays, then this is exactly equivalent to your model, except that the cytotoxicity parameter also would now include normal cell death and differentiated cell reproduction.

5. Could the kappa threshold values be reported in the main text? How much would they change if the training and testing patients were randomly reassigned?

6. This point has now been fully addressed, and the newly added references are helpful.

Minor concerns:

1. Fully addressed, thank you.

2. The newly added Fig. S5 is very useful for visualizing the system dynamics. Immediately below equation (1) in the methods, it would be useful to provide the ODE for the fraction of stem cells $s=S/(S+D)$, and to show that this ODE remains the same when cell death and differentiated cell reproduction are incorporated, as mentioned above.

3. Fully addressed. The model sensitivity analysis in Fig. S2 is quite nice, and Fig. S3 is a great

addition.

4. It would be useful to include a caption for Table S1. It is strange how much larger the MSE is for 2 of the 4 models, leading me to wonder if the following models would be more useful choices: (i) $\alpha = A - Bp$, (ii) $\log(\alpha) = A - Bp$, (iii) $\log(\alpha) = A - B\log(p)$, and (iv) $\alpha = A - B\log(p)$. Of course, (i) is already in the table assuming that B can be negative; (ii) is equivalent to the exponential model already in the table; (iii) captures the remaining two models in the table with the assumption that $\alpha \rightarrow 0$ as p increases; and (iv) is not yet present in the table but would be useful to consider. Also, is there a reason for the current table layout, or could it be rearranged to have 4 rows and 2 columns?

5. Addressed, thank you.

6. Fully addressed. The updates to the figures are all very helpful.

Reviewer #2 (Remarks to the Author):

General Comments:

The modifications by the authors greatly enhanced the manuscript however there are some further clarifications that are needed. "The overall hypothesis is in patients with hormone sensitive disease, PSA dynamics can help identify patients that will be enriched for PCaSC during treatment as a plausible mechanism of resistance evolution. They concluded that model simulations based on PSA dynamics from first IADT cycle could identify patients who would benefit from concurrent docetaxel". This is reasonable and clinically viable. However, what this model does not explain well is the clinical data in patients with metastatic hormone sensitive disease. While the authors, reviewed the data with androgen deprivation therapy (ADT) and docetaxel showing a benefit in men with metastatic hormone sensitive; there is significant support for the use of combined AR directed therapy (abiraterone acetate and now enzalutamide) with ADT which shows similar improvement in outcomes. More concerning is that data actually shows a significant benefit in those with low volume disease when docetaxel did not. Thus, if the authors are hypothesizing that chemotherapy has an effect on the PCaSC (presumed to be AR resistant) yet in the same population of patients, abiraterone and enzalutamide has the same effect (working through the AR pathway); how do the authors rectify these clinical findings? May consider better defining PCaSC as resistance cells and that PSA dynamics may define a population which need additional therapy not just chemotherapy. Please also note that in adjuvant and neo-adjuvant studies with ADT +/- docetaxel did not make a consistent significant difference in outcomes in the current completed studies.

The authors conclude that IADT outcomes in prior studies were adversely affected by the 8 month induction period however these patients were very advanced metastatic prostate cancer patients. Since the model was calibrated and validated on patients with predominantly stage T2b and T3 disease, with Gleason scores ranging from 2-9 can this data be extrapolated to patients with much larger volume of disease? Do we know that the PSA dynamics are the same for low and high volume disease? Clinically there appears to be difference in PSA dynamics. Do you have PSA dynamic data on patients with metastatic disease?

Specific comments:

1. Abstract\conclusions need to be rectified based on what we current know clinically.
2. Page 9; In the ongoing pilot trial using intermittent abiraterone in men with castrate resistant prostate cancer is not relevant here and confusing to the readers since all other data is based in patients with hormone sensitive prostate cancer. Would remove any discussion of intermittent therapy in CRPC.

We thank the reviewers for reviewing our manuscript revisions. We greatly appreciate all of the reviewers' comments and suggestions. Please find our point-by-point response to each of the concerns below.

Reviewers' comments:

Reviewer #1 (Remarks to the Author):

Thank you to the authors for taking the time to address many of our initial concerns. The paper has been improved during revision, and a few of our concerns still remain as outlined below.

Major concerns:

1. An open question remains to what extent this paper's conclusions might change if the random assignment of patients to the training and testing groups had been different. If possible, could the authors either (a) split the patients again randomly to obtain new training and testing sets or (b) switch the patients in the current training and testing sets, in order to ensure that the results are not sensitive to this random assignment?

Response: *Thank you for your comment. To test whether or not our results are dependent on the random assignment of the patients, we re-randomized the data five times (including switching the original training and testing cohorts). We computed the 95% confidence intervals for the uniform optimal parameters φ and ρ , as well as the overall accuracy and included them in Table S2. From these, we can see that the model is able to accurately predict the response with high accuracy regardless of the data randomization.*

To better account for under-represented resistant patients in the training cohort, we also completed a leave-one-out study on our set of 70 patients. This resulted in a significantly higher sensitivity (increased from 57% to 73%) and our overall accuracy remained high at 89% (Table S3). We have added these results to the revised manuscript and amended the discussion of our work accordingly (page 7, paragraph 3 & page 12). The leave-one-out study resulted in relatively narrow confidence intervals around our uniform parameters φ and ρ . Consequently, we repeated the leave-one-out study using the mean values for φ and ρ . This resulted in a significantly lower sensitivity (40%), suggesting that the accuracy of our predictions is dependent upon the values of these parameters, as well as the threshold values chosen. These results were not included in the revised manuscript.

2. Because the authors were unable to find parameter values from the literature, it would be useful to include values for at least the self-renewal probability of other types of stem cells, which have been inferred by several papers directly from division data. Also, it seems that the 95% intervals reported are interquartile ranges rather than confidence intervals. These are also useful, but I think it would be even more important to report the confidence interval for each inferred parameter. For example, with 95% confidence can you conclude that the PSA decay rate falls into some interval that you can report? Currently only the best estimate $\phi = 0.0856$ is reported. Likewise, can you report a confidence interval for the PCaSC self-renewal rate of patient 091, for example? Currently only the best estimate $pS = 0.0201$ is reported; a confidence interval would help address whether patient heterogeneity in pS and α is due to inference uncertainty or real patient differences.

Response: *We agree with the reviewer that such probabilities have been reported in the literature. We do, however, have sincere concerns about comparing net rates in our model to experimentally derived parameters. We extended our discussion in the first revision of the manuscript to include the following paragraph:*

“The mathematical model was developed with the intent to make clinically actionable predictions. To balance model complexity and prediction accuracy, the presented simple model has been derived (31). While numerous additional biological mechanisms are contributing to the observed PSA dynamics, it will become impossible to calibrate and validate their contributions from the measured clinical data. For reproducibility, we report the trained parameter values for the presented model. However, modification of the functional form of the developed model terms or inclusion of additional mechanisms will change each parameter. Therefore, herein estimated parameter values should not be taken as biological ground truth, nor are they translatable to different mathematical models.”

As the actual parameter values are highly dependent on the functional form of the mathematical model, experimentally derived rates are also very specific to the chosen cell line, experimental condition, etc. Therefore, it is our opinion that it will be misleading to compare our derived model-dependent parameters to other experimental condition-dependent parameters from different types of stem cells. The purpose of the model and presented work is to make predictions, not to identify abstract biological rates.

We confirmed that the intervals given in the results section are indeed the 95% confidence intervals. As the optimizer converged to one parameter pair (p_s, α) for each patient, we do not have individual confidence intervals for each patient. However, we included the 95% confidence intervals over all of the patients in the training cohort for the patient-specific parameters p_s and α in the initial submission (page 5, paragraph 1 & page 6, paragraph 1).

3. This concern has been fully addressed.

4. A reader might be concerned that the ODE model would behave quite differently if cell death or the reproduction of differentiated cells were to be included. However, perhaps you can provide the following argument for why they can both be neglected: If stem cells die with rate d_1 , and if differentiated cells die with rate d_2 and reproduce with rate r , then the dS/dt equation would include an additional $-d_1 * S$ term, and the dD/dt equation would include an additional $+(r - d_2)D$ term. But then the ODE describing the fraction of stem cells $s = S/(S+D)$ would be $ds/dt = (aT_x - r - d_1 + d_2)s(1-s) - (1-p)*\lambda s^2$. If you assume that $r = d_1 - d_2$ during treatment holidays, then this is exactly equivalent to your model, except that the cytotoxicity parameter also would now include normal cell death and differentiated cell reproduction.

Response: *We agree with the reviewer that the ODE model would behave differently if different biological terms or mechanisms had been included. We did not develop all possible or the most biologically accurate model, but a model that has plausible biology to make accurate predictions to help guide clinical decision making. We can explore multiple additional models that include additional mechanisms; a thorough analysis of these is beyond the purpose of this paper. We identified a plausible model that mimics clinically observed data and makes actionable predictions with unprecedented high confidence and accuracy.*

We agree that the analysis of the stem cell fraction is a nice addition to the manuscript and have provided this in the first revision. We respectfully disagree with the suggestion to include a discussion of additional terms that can then lead to a simplification that becomes

equivalent to the results we already presented. A major concern is that the reviewer-proposed assumption $r=d_1-d_2$ has little biological motivation. It is unclear why the rate at which differentiated cells reproduce equals the death rate of differentiated cells subtracted from the death rate of stem cells. Thus, while mathematically interesting, we argue to refrain from inclusion for manuscript clarity.

5. Could the kappa threshold values be reported in the main text? How much would they change if the training and testing patients were randomly reassigned?

Response: *We have moved the threshold discussion from the Materials and Methods into the Results section (page 7, paragraph 1). We also re-randomized the patients into training and testing cohorts five times and included the confidence intervals for φ and ρ , as well as the thresholds κ_i (Table S2). We also completed a leave-one-out study for all 70 patients, obtaining threshold values for each individual training cohort (95% confidence intervals in Table S3). The thresholds changed depending on training set, but the overall confidence intervals were narrow. Confidence intervals for the fourth cycle threshold are noticeably wide due to few patients developing resistance during fourth cycle. As the confidence intervals are relatively narrow, we have also attempted to identify a uniform threshold for each cycle. This led to lower sensitivities, showing that variation in the thresholds are required for the hurricane-style prediction model. We have extended the discussion in the revised manuscript (page 14, paragraph 2).*

6. This point has now been fully addressed, and the newly added references are helpful.

Minor concerns:

1. Fully addressed, thank you.

2. The newly added Fig. S5 is very useful for visualizing the system dynamics. Immediately below equation (1) in the methods, it would be useful to provide the ODE for the fraction of stem cells $s=S/(S+D)$, and to show that this ODE remains the same when cell death and differentiated cell reproduction are incorporated, as mentioned above.

Response: *Thank you for the suggestion. As we do not actually use the stem cell proportion for the model forecasting, we believe it is best not to include the equation in the main portion of the text. Instead, we opted to including it in the caption of Figure S5.*

3. Fully addressed. The model sensitivity analysis in Fig. S2 is quite nice, and Fig. S3 is a great addition.

Response: *Thank you. We agree that it adds to the manuscript.*

4. It would be useful to include a caption for Table S1. It is strange how much larger the MSE is for 2 of the 4 models, leading me to wonder if the following models would be more useful choices: (i) $\alpha = A - Bp$, (ii) $\log(\alpha) = A - Bp$, (iii) $\log(\alpha) = A - B\log(p)$, and (iv) $\alpha = A - B\log(p)$. Of course, (i) is already in the table assuming that B can be negative; (ii) is equivalent to the exponential model already in the table; (iii) captures the remaining two models in the table with the assumption that $\alpha \rightarrow 0$ as p increases; and (iv) is not yet present in the table but would be useful to consider. Also, is there a reason for the current table layout, or could it be rearranged to have 4 rows and 2 columns?

Response: Thank you for the suggestion. A caption has been added to the table. The large MSE in two of the four models is due to the form of the equations. As shown in Fig. 2, p_s can be small (i.e. $< 1e-10$), resulting in a very large MSE. The equation $\alpha = A - B \log(p_s)$ has been added to the analysis. MSE is reported in Table S1. While multiple functional forms exist, with similar fits to the data (differing by less than 10^{-3} in most cases), we chose the exponential relationship as it has been shown to make clinically actionable predictions with high accuracy.

5. Addressed, thank you.

6. Fully addressed. The updates to the figures are all very helpful.

Reviewer #2 (Remarks to the Author):

General Comments:

The modifications by the authors greatly enhanced the manuscript however there are some further clarifications that are needed. The overall hypothesis is in patients with hormone sensitive disease, PSA dynamics can help identify patients that will be enriched for PCaSC during treatment as a plausible mechanism of resistance evolution. They concluded that model simulations based on PSA dynamics from first IADT cycle could identify patients who would benefit from concurrent docetaxel. This is reasonable and clinically viable. However, what this model does not explain well is the clinical data in patients with metastatic hormone sensitive disease. While the authors, reviewed the data with androgen deprivation therapy (ADT) and docetaxel showing a benefit in men with metastatic hormone sensitive; there is significant support for the use of combined AR directed therapy (abiraterone acetate and now enzalutamide) with ADT which shows similar improvement in outcomes. More concerning is that data actually shows a significant benefit in those with low volume disease when docetaxel did not. Thus, if the authors are hypothesizing that chemotherapy has an effect on the PCaSC (presumed to be AR resistant) yet in the same population of patients, abiraterone and enzalutamide has the same effect (working through the AR pathway); how do the authors rectify these clinical findings?

Response: We agree with the reviewer that the model may not explain clinical data in patients with metastatic disease. The model has been built using data of patients with biochemically recurrent disease, and should only be evaluated for this population. Once we have data available on abiraterone acetate and enzalutamide responses, the model will be readily available to be trained on these data as well.

May consider better defining PCaSC as resistance cells and that PSA dynamics may define a population which need additional therapy not just chemotherapy. Please also note that in adjuvant and neo-adjuvant studies with ADT +/- docetaxel did not make a consistent significant difference in outcomes in the current completed studies.

Response: We have added discussion of more current findings comparing ADT alone with abiraterone and ADT and emphasized that PCaSCs are resistant to ADT (page 10, paragraph 2). Furthermore, the presented results corroborate these findings that ADT +/- docetaxel did not make a consistent significant difference in outcomes. What our results demonstrate is that despite the lack of clear benefit on the population as a whole, we may be able to identify selected

patients who may benefit from concurrent docetaxel and, perhaps more importantly, who does not. We have added this sentence to the revised discussion (page 13, paragraph 2).

The authors conclude that IADT outcomes in prior studies were adversely affected by the 8 month induction period however these patients were very advanced metastatic prostate cancer patients. Since the model was calibrated and validated on patients with predominantly stage T2b and T3 disease, with Gleason scores ranging from 2-9 can this data be extrapolated to patients with much larger volume of disease? Do we know that the PSA dynamics are the same for low and high volume disease? Clinically there appears to be difference in PSA dynamics. Do you have PSA dynamic data on patients with metastatic disease?

Response: *The induction period has been used in both localized and metastatic prostate cancer. The referenced study by Crook et al. compared IADT to continuous ADT in PCa patients with localized disease, while the study by Hussein et al. was conducted in the metastatic setting. In the more localized setting, there is even more justification not to include such an extended induction period due to these patients having such low (or no) volume disease. The resulting competitive release of the more resistant cell phenotype during IADT will likely result in resistance developing at a rate comparable to continuous ADT. Nevertheless, the cited studies provided clear motivation to investigate forfeiting the induction period. At this time, we do not know if the PSA dynamics are the same for low- and high-volume disease and do not feel comfortable speculating. Unfortunately, we do not have PSA dynamic data on patients with metastatic disease to make a concrete conclusion.*

Specific comments:

1. Abstract\conclusions need to be rectified based on what we current know clinically.

Response: *We have added discussion of more current findings from studies investigating the effects of concurrent taxane and antiandrogen treatments with ADT in the metastatic setting.*

2. Page 9; In the ongoing pilot trial using intermittent abiraterone in men with castrate resistant prostate cancer is not relevant here and confusing to the readers since all other data is based in patients with hormone sensitive prostate cancer. Would remove any discussion of intermittent therapy in CRPC.

Response: *Thank you for bringing this to our attention. We have removed the sentences mentioning the pilot trial for clarity.*

REVIEWERS' COMMENTS:

Reviewer #1 (Remarks to the Author):

Thank you to the authors for quickly and fully addressing all of my comments. I have no remaining concerns.

Reviewer #2 (Remarks to the Author):

Since the model was calibrated and validated on patients with predominantly stage T2b and T3 disease, with Gleason scores ranging from 2-9; a statement in the conclusion should clarify: "That this model has only been validated in a small cohort of patients with stage T2b and T3 disease, with Gleason scores ranging from 2-9 and further work in patients with more advanced or metastatic prostate cancer is needed."

All other responses to the critique are acceptable.

We thank the reviewers for reviewing our manuscript revisions. We greatly appreciate all of the reviewers' comments and suggestions. Please find our point-by-point response to each of the concerns below.

Reviewers' comments:

Reviewer #1 (Remarks to the Author):

Thank you to the authors for quickly and fully addressing all of my comments. I have no remaining concerns.

Response: *Thank you for taking the time to review our work.*

Reviewer #2 (Remarks to the Author):

Since the model was calibrated and validated on patients with predominantly stage T2b and T3 disease, with Gleason scores ranging from 2-9; a statement in the conclusion should clarify: "That this model has only been validated in a small cohort of patients with stage T2b and T3 disease, with Gleason scores ranging from 2-9 and further work in patients with more advanced or metastatic prostate cancer is needed."

Response: *The text has been updated accordingly (page 15, paragraph 2).*

All other responses to the critique are acceptable.

Response: *Thank you.*